# Clinical and genetic associations of deep learning-derived cardiac magnetic resonance-based left ventricular mass

Shaan Khurshid [1,2,3], Julieta Lazarte [1,2,4], James P. Pirruccello [1,2,5], Lu-Chen Weng [1,2], Seung Hoan Choi [1,2], Amelia W. Hall[6], Xin Wang[1,2], Samuel F. Friedman [7], Victor Nauffal[8], Kiran J. Biddinger[1,2], Krishna G. Aragam[1,2,5], Puneet Batra[7], Jennifer E. Ho [2,9], Anthony A. Philippakis [7], Patrick T. Ellinor [1,2,3] & Steven A. Lubitz [1,2,3] ✉

Left ventricular mass is a risk marker for cardiovascular events, and may indicate an underlying cardiomyopathy. Cardiac magnetic resonance is the gold-standard for left ventricular mass estimation, but is challenging to obtain at scale. Here, we use deep learning to enable genome-wide association study of cardiac magnetic resonance-derived left ventricular mass indexed to body surface area within 43,230 UK Biobank participants. We identify 12 genome-wide associations (1 known at *TTN* and 11 novel for left ventricular mass), implicating genes previously associated with cardiac contractility and cardiomyopathy. Cardiac magnetic resonance-derived indexed left ventricular mass is associated with incident dilated and hypertrophic cardiomyopathies, and implantable cardioverter-defibrillator implant. An indexed left ventricular mass polygenic risk score ≥90th percentile is also associated with incident implantable cardioverter-defibrillator implant in separate UK Biobank (hazard ratio 1.22, 95% CI 1.05-1.44) and Mass General Brigham (hazard ratio 1.75, 95% CI 1.12-2.74) samples. Here, we perform a genome-wide association study of cardiac magnetic resonance-derived indexed left ventricular mass to identify 11 novel variants and demonstrate that cardiac magnetic resonance-derived and genetically predicted indexed left ventricular mass are associated with incident cardiomyopathy.

Left ventricular hypertrophy (LVH) is defined as pathologically increased left ventricular mass (LVM)[1] and is associated with increased risk of cardiovascular events including heart failure (HF)[1–3], stroke[1], atrial fibrillation (AF)[4], and sudden cardiac death[5]. Increased LVM is also a hallmark of certain primary cardiomyopathies such as hypertrophic cardiomyopathy (HCM) and some dilated cardiomyopathies (DCM). Although LVM can be estimated using 12 lead electrocardiograms or echocardiography, cardiac magnetic resonance (CMR)

[1]Cardiovascular Research Center, Massachusetts General Hospital, Boston, MA, USA. [2]Cardiovascular Disease Initiative, Broad Institute of Harvard and the Massachusetts Institute of Technology, Cambridge, MA, USA. [3]Demoulas Center for Cardiac Arrhythmias, Massachusetts General Hospital, Boston, MA, USA. [4]Department of Medicine, Schulich School of Medicine and Dentistry, Western University, London, ON, Canada. [5]Division of Cardiology, Massachusetts General Hospital, Boston, MA, USA. [6]Gene Regulation Observatory, Broad Institute of Harvard and the Massachusetts Institute of Technology, Cambridge, MA, USA. [7]Data Sciences Platform, Broad Institute of Harvard and the Massachusetts Institute of Technology, Cambridge, MA, USA. [8]Division of Cardiology, Brigham and Women's Hospital, Boston, MA, USA. [9]CardioVascular Institute and Division of Cardiology, Department of Medicine, Beth Israel Deaconess Medical Center, Boston, MA, USA. ✉e-mail: slubitz@mgh.harvard.edu

 1

**Fig. 1 | Overview of study design and flow.** We obtained CMR-derived LVM index in 44,375 individuals undergoing CMR imaging. We performed a genome-wide association study of CMR-derived LVMI and assessed for associations between CMR-derived LVMI and cardiovascular outcomes. Using GWAS results, we developed a polygenic risk score for LVMI, and applied it to 443,326 separate UK Biobank participants with genetic data (left), and 29,354 individuals from the independent Mass General Brigham Biobank (right), to assess for associations between genetically determined LVMI and cardiovascular outcomes.

offers more accurate and reproducible quantification, and has therefore emerged as the gold standard for diagnosing LVH[6].

Imaging-based estimation of LVM typically requires LV segmentation, which is usually performed manually and requires substantial time and expertise. As a result, genetic analyses of imaging-based LVM have been limited by modest sample sizes. Genome-wide association studies (GWAS) of echocardiography-based LVM identified a single susceptibility locus downstream of *SPCS3*[7–9]. More recently, a genome-wide association study within 19,000 individuals[10] identified significant variants in the gene *TTN* associated with CMR-based LVM.

Here, we apply a validated deep learning approach to automate estimation of LVM using CMR images (Machine Learning for Health – Segmentation [ML4H$_{seg}$]), to maximize power to detect genetic associations underlying CMR-derived LVM[11]. Specifically, we implement ML4H$_{seg}$ to estimate LVM using CMRs from nearly 50,000 participants in the UK Biobank. Given body size is a major determinant of LV size and mass[12], we analyze LVMI (i.e., LVM indexed by body surface area) in our primary analyses, and assess unindexed LVM in secondary analyses. Our GWAS of LVMI identifies 12 independent variants meeting genome-wide significance, including 11 novel associations. Using expression quantitative trait loci (eQTLs), transcriptome-wide association testing (TWAS), and tissue-specific expression levels, we propose several candidate genes, many of which have been previously associated with cardiac contractility and cardiomyopathy. We additionally develop a polygenic risk score (PRS) for LVMI, and demonstrate that both phenotypic and genetic LVMI are associated with incident cardiovascular diseases including cardiomyopathy.

## Results

### Genome-wide association study of CMR-derived LVM

We conducted a multi-ancestry GWAS including 43,230 individuals (91% European ancestry) (Fig. 1, Supplementary Table 1). The analysis included 9.9 million common variants imputed at an INFO score ≥0.30 and having minor allele frequency (MAF) ≥1%. The genomic control factor was 1.15 with a linkage disequilibrium score regression intercept of 1.00, consistent with polygenicity of the LVMI trait as opposed to inflation (Supplementary Fig. 1). Observed scale $h^2$ for LVMI was 0.26 (standard error [SE] 0.02).

The GWAS initially revealed 12 candidate SNPs associated with CMR-derived LVMI at genome-wide significance (Table 1 and Fig. 2). Conditional analyses identified an additional variant on chromosome 2, and that the two variants on chromosome 17 located 914 kb apart ($r^2 = 0.37$) were not independent, ultimately resulting in 12 lead SNPs for LVMI. The SNP most strongly associated with LVMI (rs2255167, $p = 1.4 \times 10^{-26}$) was located at the *TTN* locus on chromosome 2 and has been previously associated with LVM. *TTN* is highly expressed in LV tissue (Supplementary Table 2)[10]. The remaining loci ($n = 11$) were novel, with many located at or proximate to genes implicated in arrhythmias, cardiomyopathy and cardiomyocyte function, including *FLNC*, *MYOZ1*, *MAPT*, *WNT*, *CLCN6*, *MYBPC3* and *SYNPO2L*. Regional association plots for each genome-wide significant SNP are shown in Supplementary Fig. 2. Results for 18 additional variants having suggestive but not genome-wide significant associations are shown in Supplementary Table 3. A secondary GWAS of unindexed LVM revealed 12 genome-wide significant SNPs, of which 6 overlapped with the primary LVMI GWAS, and a 7th was a strong proxy ($r^2 = 0.87$). Loci unique to analyses of unindexed LVM appeared primarily enriched for genes associated with body size (e.g., *FTO*, *HMGA2*, *GDF5*), although *FTO* has also been implicated in HF[13] and *CDKN1A* has been associated with DCM in a recent multi-trait analysis[14] (Supplementary Table 4 and Supplementary Fig. 3).

In GWAS restricted to individuals of European ancestry, 14 loci met genome-wide significance, of which 12 were either a lead variant or a strong proxy ($r^2 > 0.8$) for a lead variant in the primary GWAS (Supplementary Table 5 and Supplementary Figs. 4 and 5). The two loci unique to the European ancestry analysis were rs143973349, an insertion-deletion variant located near *FLNC*, a gene highly expressed in LV tissue and previously associated with familial hypertrophic, restrictive, and arrhythmogenic cardiomyopathies, and rs142032045, located in a gene-rich region closest to *DOC2A* and near several variants previously associated with body size[15–18]. The variant near *FLNC* had a suggestive association with LVMI in the primary multi-ancestry GWAS, while the variant near *DOC2A* did not ($p = 3.2 \times 10^{-7}$ and $p = 1.1 \times 10^{-5}$, respectively). The only variant meeting genome-wide significance in the primary mixed-ancestry GWAS that was not a lead variant in the

**Table 1 | Variants associated with CMR-derived left ventricular mass index in the mixed-ancestry GWAS**

| rsID | Chr | Position (hg38) | Closest gene(s) | Function | Risk/alt allele | RAF | Beta | SE | P value* |
|---|---|---|---|---|---|---|---|---|---|
| rs143800963 | 1 | 11835418 | *CLCN6* | Intronic | C/A | 0.95 | 0.95 | 0.16 | $4.2 \times 10^{-9}$ |
| rs2255167[†] | 2 | 178693555 | *TTN* | Intronic | T/A | 0.81 | 0.97 | 0.09 | $3.2 \times 10^{-26}$ |
| rs10497529[‡] | 2 | 178975161 | *CCDC141* | Missense | G/A | 0.96 | 1.28 | 0.20 | $2.2 \times 10^{-9}$ |
| - | 5 | 133066736 | *HSPA4* | Indel | CTT/C | 0.72 | 0.50 | 0.08 | $1.6 \times 10^{-9}$ |
| rs9388498 | 6 | 126552277 | *CENPW* | - | G/T | 0.81 | −0.55 | 0.10 | $4.1 \times 10^{-9}$ |
| rs34163229 | 10 | 73647154 | *SYNPO2L* | Missense | G/T | 0.86 | −0.60 | 0.10 | $1.0 \times 10^{-8}$ |
| rs3729989 | 11 | 47348490 | *MYBPC3* | Missense | T/C | 0.87 | −0.61 | 0.11 | $1.8 \times 10^{-8}$ |
| rs28552516 | 12 | 121592356 | *KDM2B* | Intronic | C/T | 0.85 | −0.58 | 0.10 | $1.5 \times 10^{-8}$ |
| rs6598541 | 15 | 98727906 | *IGF1R* | Intronic | A/G | 0.36 | −0.42 | 0.08 | $4.6 \times 10^{-8}$ |
| rs56252725 | 16 | 14995819 | *PDXDC1* | Intronic | G/A | 0.75 | 0.54 | 0.09 | $3.7 \times 10^{-9}$ |
| rs6503451 | 17 | 45870981 | *MAPT* | Intronic | T/C | 0.67 | −0.52 | 0.08 | $1.1 \times 10^{-10}$ |
| rs199501[§] | 17 | 46785247 | *WNT3* | Intronic | A/G | 0.24 | 0.55 | 0.09 | $1.1 \times 10^{9}$ |
| rs62621197 | 19 | 8605262 | *ADAMTS10* | Missense | C/T | 0.96 | 1.11 | 0.20 | $2.9 \times 10^{-8}$ |

*Chr* chromosome, *RAF* risk allele frequency, *OR* odds ratio.
*Denotes two-sided *p* value corresponding to BOLT-LMM $\chi^2$ statistic.
[†]Locus previously reported for LVM[10].
[‡]Variant identified in conditional analysis conditioned on lead SNPs (beta, standard error, and *p* value are adjusted).
[§]Association no longer observed in analysis conditioned on rs6503451.

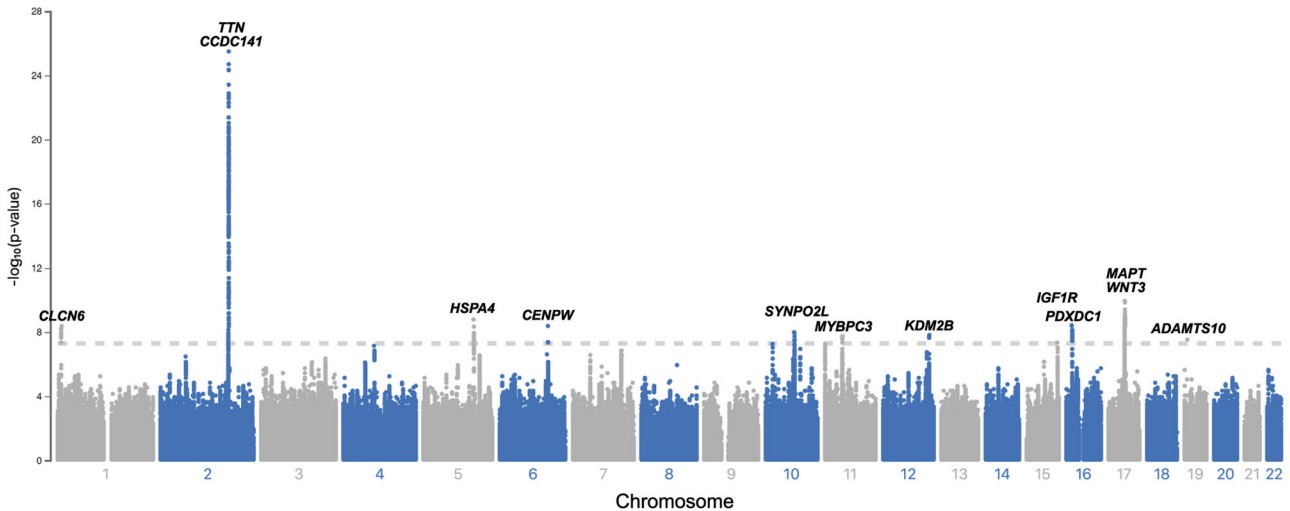

**Fig. 2 | Manhattan plot of mixed-ancestry GWAS for CMR-derived LVM index.**
Depicted across increasing chromosome (x-axis) are the association results of the primary mixed-ancestry GWAS of left ventricular mass index. The y-axis plots the negative $\log_{10}$ of the two-sided *p* value corresponding to BOLT-LMM $\chi^2$ statistic. Variants meeting the standard multiplicity correction for genome-wide significance ($p < 5 \times 10^{-8}$, depicted by hashed horizontal line), are labeled by the closest gene to the lead variant.

European-only GWAS did have a suggestive association (rs6598541 near *IGF1R* $p = 7.7 \times 10^{-8}$).

Results of secondary GWAS analyses, including rank-based inverse normal transformed LVMI, LVMI indexed using the 2.7th power of height, LVMI indexed using lean body mass, LVMI with exclusions for prevalent myocardial infarction and heart failure, and unindexed LVM adjusted for height and weight, are shown in Supplementary Tables 6-10. Results obtained using alternative indexing methods were broadly consistent with the primary analysis in terms of variants identified and effect directions. A summary of association results for the lead variants identified in the primary GWAS tested across varying indexing methods is shown in Supplementary Table 11.

**Bioinformatics and in silico functional analyses to determine candidate genes**

In total, of the 12 independent lead SNPs, eight (or their proxies at $r^2 \geq 0.8$) were significant eQTLs in LV and/or AA tissue samples (Fig. 3). The locus including variant rs143973349 unique to the European ancestry analysis also included eQTLs for LV and AA tissue. For a significant proportion of candidate genes, expression was identified in both LV and AA tissue samples. We then performed TWAS and identified 6 genes across 5 loci where predicted expression was associated with LVMI. Each of the genes implicated by TWAS was also an eQTL for either LV or AA (Fig. 3). Using Hi-C analysis, we observed several potentially relevant chromatin interactions, including between lead variant rs56252725 on chromosome 16 and gene *MYH11*, which encodes an isoform of the myosin heavy chain which is highly expressed in LV tissue and has been associated with electrocardiogram amplitude, and between lead variant rs143973349 (European-only analysis) and gene *CCDC136*, which encodes a membrane protein and in which variants have been previously associated with dilated and hypertrophic cardiomyopathies. Detailed results of eQTL, TWAS, and Hi-C analyses are shown in Supplementary Table 2.

Probable candidate genes at each locus of interest are summarized in Fig. 3. In several cases, the closest gene was additionally supported by either eQTL or TWAS prioritization, including *SYNPO2L* near

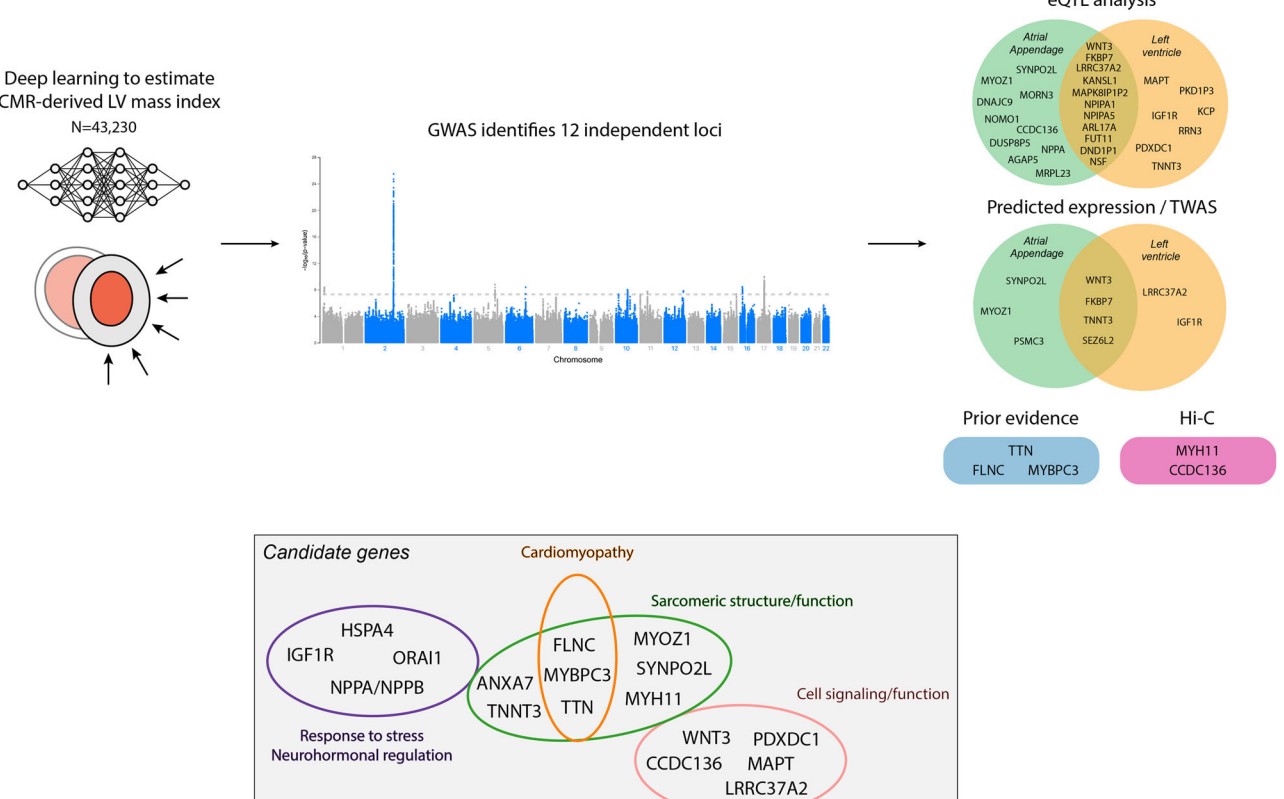

**Fig. 3 | Candidate gene summary.** Depicted is a summary of study results. We used a deep learning algorithm to perform a GWAS of CMR-derived LVMI in 43,230 individuals, finding 12 independent loci associated with LVMI. Using proximity to lead variants, expression quantitative trait locus (eQTL) analysis, transcriptome-wide association studies (TWAS), Hi-C analysis, LV tissue-specific expression levels, and prior evidence, we identified candidate genes across the 12 loci. Candidate genes were enriched for genes involved in stress response and neurohormonal regulation, cardiac structure and cardiomyopathy, and cell signaling/function (gray box).

rs56252725, *IGF1R* near rs6598541, *PDXDC1* near rs56252725, *MAPT* near rs6598541, and *WNT3* near rs199501. In selected instances, downstream analyses prioritized alternative genes, including *NPPA* near rs143800963 and *ORAI1* near rs28552516, with both genes having substantial expression in LV tissue. Selected genes prioritized based on strong biologic plausibility or previous associations with LVM included *TTN* near rs255167, *MYBPC3* near rs3729989, and *FLNC* near rs143973349 (EUR only subset). *TTN*, *MYBPC3*, and *FLNC* are also substantially expressed in LV tissue (Supplementary Table 2).

**Comparison to prior associations with LV measurements and cardiovascular traits**

We assessed whether the significant loci we identified have been previously associated with LV measurements[10,19] and cardiovascular traits. Including the European-only analysis, a total of 4 loci have been previously associated with LV measurements. Variant rs2255167 is located on a region of *TTN* previously associated with LV mass, LV end diastolic volume, LV end-systolic volume, and LV ejection fraction. Variants rs6503451 near *MAPT* and rs199501 near *WNT3* are located at regions previously associated with LV end-systolic volume. In the European-only analysis, variant rs143973349 near *FLNC* is at a locus previously associated with LV end-systolic volume and LV ejection fraction. Several additional loci have been implicated in other cardiovascular diseases such as heart failure (e.g., rs34163229 near *SYNPO2L*), cardiomyopathy (e.g., rs2255167 near *TTN*, rs3729989 near *MYBPC3*, rs143973349 near *FLNC*), and atrial fibrillation (e.g., rs6598541 near *IGF1R*), while others have been associated with cardiovascular risk factors such as blood pressure or diabetes. Several variants are located at regions previously

associated with electrocardiographic traits such as PR interval (e.g., rs56252725 near *PDXDC1*), QRS duration (rs6598541 near *IGF1R*), and QRS amplitude (rs6503451 near *MAPT*). Variants rs28552516 near *KDM2B*, rs62621197 near *ADAMTS10*, and rs142032045 near *DOC2A* in the European-only analysis have not been previously associated with either LV or other cardiovascular traits. A summary of lead variants and their prior associations is shown in Supplementary Table 12.

**Associations between LVMI and cardiomyopathy**

We assessed for associations between CMR-derived LVMI and incident cardiovascular disease. At a median follow-up of 2.7 years (Q1:1.9, Q3:4.1), greater LVMI was consistently associated with greater risk of multiple conditions, including AF, MI, HF, DCM, HCM, and ICD implant (Supplementary Table 13). CMR-derived LVH was strongly associated with incident DCM (HR 10.9, 95% CI 4.67–20.2), HCM (HR 9.26, 95% CI 3.20–26.8), and ICD implant (HR 8.42, 95% CI 3.82–18.6). Cumulative risk of events stratified by presence versus absence of CMR-derived LVH is depicted in Fig. 4.

We next evaluated associations between LVMI genetic risk and incident outcomes. In a set of UK Biobank participants separate from the GWAS sample (*n* = 443,326), a greater LVMI PRS was associated with higher risk of multiple incident conditions including AF, HF, ventricular arrhythmias, DCM, and ICD implant (Table 2). In the independent MGB sample (*n* = 29,354), the LVMI PRS was again associated with incident ICD implant, along with suggestive associations with HCM and DCM (Table 2). In models of incident ICD risk, the relative hazard of ICD was consistently greatest at the highest levels of CMR-derived LVMI as well as LVMI PRS, with similar effect sizes in both the

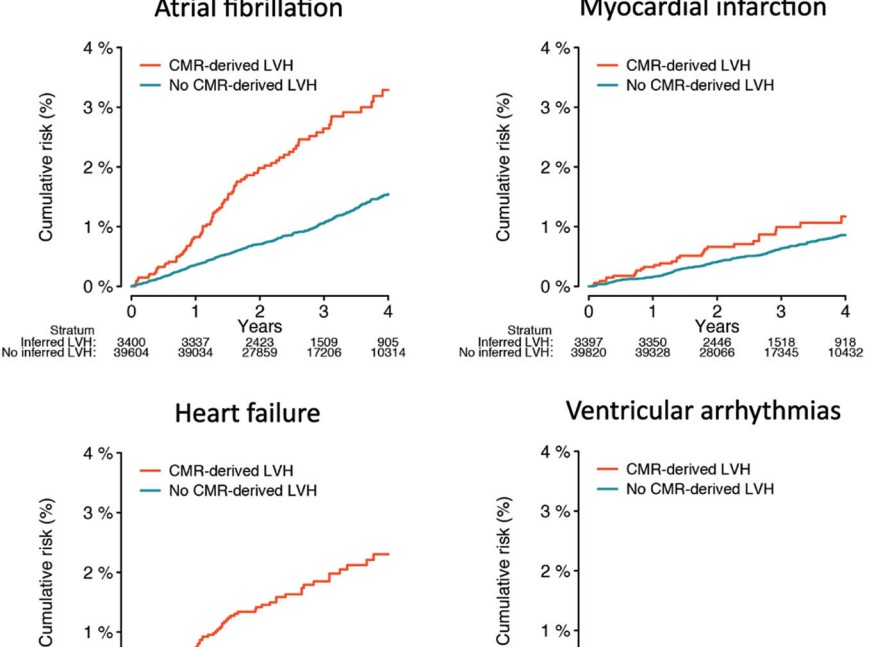

**Fig. 4 | Kaplan–Meier plots of the association between CMR-derived LVH and incident cardiovascular disease.** Plots depicting the cumulative risk of atrial fibrillation (top left), heart failure (top right), myocardial infarction (bottom left), and ventricular arrhythmias (bottom right), stratified by the presence (orange) versus absence (teal) of CMR-derived LVH. LVH was defined as LVM index (LVMI) > 72 g/m$^2$ in men and >55 g/m$^2$ in women[44]. The number at risk within each stratum over time is depicted below each plot.

UK Biobank and MGB (Fig. 5). Disease association results were generally similar in analyses restricted to individuals of European ancestry (Supplementary Table 14), and when utilizing a PRS derived from GWAS performed after exclusion of individuals with prevalent myocardial infarction and heart failure (Supplementary Table 15).

**Mendelian-randomization analyses of blood pressure and diabetes**

To assess for potential causal associations between blood pressure and CMR-derived LVMI, we performed MR analyses using genetic instruments for SBP and DBP among individuals of European ancestry. We performed analogous analyses for diabetes. In an inverse-variance weighted two-sample MR, a 1-SD increase in genetically mediated SBP was associated with a 0.27 g/m$^2$ increase in CMR-derived LVMI (95% CI 0.23–0.31, $p = 1.75 \times 10^{-41}$), and a 1-SD increase in genetically mediated DBP was associated with a 0.32 g/m$^2$ increase in CMR-derived LVMI (95% CI, 0.25–0.39, $p = 1.64 \times 10^{-20}$). A 1-SD increase in genetically mediated risk of diabetes was associated with a 0.31 g/m$^2$ increase in CMR-derived LVMI (95% CI, 0.05–0.56, $p = 0.018$). Weighted median and MR-Egger analyses demonstrated similar results for SBP and DBP, but associations with diabetes were no longer significant (weighted median: 0.19 g/m$^2$, 95% CI −0.15 to 0.53, $p = 0.26$; MR-Egger: 0.15 g/m$^2$, 95% CI −0.36 to 0.66, $p = 0.56$). MR-Egger analyses suggested no substantive directional pleiotropy in the SBP, DBP, and diabetes instruments (intercept 0.01, p-0.38 for SBP; intercept −0.02, $p = 0.04$ for DBP; intercept=0.01, $p = 0.50$ for diabetes). MR results were similar using unindexed LVM (Supplementary Table 16). MR plots are shown in Supplementary Fig. 6.

## Discussion

In the current study, we utilized a deep learning segmentation algorithm to perform GWAS of CMR-derived LVMI in nearly 50,000 individuals. Leveraging favorable statistical power and a rich imaging-based phenotype, we identified 12 independent loci associated with LVMI at genome-wide significance. Of the loci identified, 11 are novel for LV mass, 9 have not been previously associated with any LV measurement, and 2 have not been associated with any cardiovascular trait or risk factor. A European-only analysis revealed 2 additional loci which are novel for LV mass. Downstream analyses prioritize several candidate genes, including multiple genes previously associated with cardiac structure and function, as well as cardiomyopathy. Importantly, CMR-derived and genetically determined LVMI were each associated with greater risk of incident cardiovascular events, including incident, DCM, and ICD implant.

Our analyses suggest that common variants in cardiac structural and functional genes appear to be important determinants of LVM. CMR-derived LVMI was strongly associated with variation at rs2255167, located within the gene encoding the large sarcomeric protein titin and previously associated with LV mass[10], as well as LV volumes and ejection fraction[19]. *MYOZ1*, which encodes a sarcomeric protein involved in calcineurin signaling and was prioritized by both eQTL and TWAS analysis, has been previously associated with HF[13] and AF[20]. A mouse knockout of *MYOZ1* resulted in increased exercise capacity through activation of the nuclear factor of activated T-cells[21]. Another gene prioritized by both eQTL and TWAS, *TNNT3*, encodes a troponin T isoform which is highly expressed in LV tissue. The *TNNT3* R63H variant has been shown to result in increased contractility in mouse skeletal muscle and is a cause of the human disease Arthrogryposis (Type 2B2)[22], characterized by limb contractures (i.e., excessive muscular contraction). *SYNPO2L*, an actin-related protein expressed in LV myocardium, has been previously associated with AF[23], HF[24], HCM[14], and voltage-duration product (a clinical indicator of LVH)[25].

Several of the candidate genes we identified prioritize neurohormonal regulation and response to physiologic stress as potential

**Table 2 | Associations between LVMI PRS and incident disease**

| | N events/N total[†] | Follow-up, yrs (Q1,Q3) | Hazard ratio for covariate (95% CI)* | | |
|---|---|---|---|---|---|
| | | | PRS (per 1 SD) | PRS (90th percentile) | PRS (95th percentile) |
| **UK Biobank** | | | | | |
| Atrial fibrillation | 25050/435917 | 11.8 (11.0,12.6) | 1.01 (1.00–1.03) | 1.03 (0.98–1.07) | 1.04 (0.98–1.10) |
| Myocardial infarction | 13405/432044 | 11.8 (11.0,12.6) | 1.03 (1.01–1.05) | 1.05 (0.99–1.11) | 1.10 (1.02–1.18) |
| Heart failure | 13540/440590 | 11.9 (11.0,12.6) | 1.04 (1.02–1.05) | 1.06 (1.00–1.12) | 1.08 (1.00–1.16) |
| Ventricular arrhythmias | 4882/442295 | 11.9 (11.1,12.6) | 1.06 (1.03–1.09) | 1.13 (1.04–1.24) | 1.17 (1.04–1.32) |
| Dilated cardiomyopathy[‡] | 1023/443013 | 11.9 (11.1,12.6) | 1.10 (1.04–1.17) | 1.15 (0.95–1.40) | 1.29 (1.00–1.66) |
| Hypertrophic cardiomyopathy[‡] | 420/443150 | 11.9 (11.1,12.6) | 1.08 (0.98–1.09) | 0.95 (0.68–1.33) | 1.23 (0.82–1.86) |
| Implantable defibrillator | 1444/443216 | 11.9 (11.1,12.6) | 1.07 (1.02–1.13) | 1.22 (1.05–1.44) | 1.22 (0.98–1.51) |
| **Mass General Brigham** | | | | | |
| Atrial fibrillation | 1332/25316 | 2.9 (2.0,4.1) | 1.01 (0.95–1.06) | 1.02 (0.85–1.22) | 1.03 (0.80–1.31) |
| Myocardial infarction | 695/25592 | 2.9 (2.0,4.1) | 0.99 (0.92–1.06) | 0.97 (0.74–1.25) | 0.71 (0.47–1.07) |
| Heart failure | 1074/25063 | 2.9 (2.0,4.1) | 0.97 (0.91–1.03) | 1.18 (0.97–1.42) | 1.00 (0.76–1.33) |
| Ventricular arrhythmias | 944/26990 | 3.0 (2.0,4.2) | 0.99 (0.93–1.05) | 1.00 (0.81–1.24) | 1.03 (0.76–1.38) |
| Dilated cardiomyopathy | 492/28821 | 3.0 (2.1,4.2) | 1.06 (0.97–1.16) | 1.27 (0.97–1.67) | 1.06 (0.70–1.59) |
| Hypertrophic cardiomyopathy | 183/28731 | 3.0 (2.1,4.2) | 1.14 (0.98–1.32) | 1.04 (0.64–1.69) | 0.82 (0.38–1.75) |
| Implantable defibrillator | 152/28454 | 3.0 (2.1,4.2) | 1.05 (0.89–1.24) | 1.75 (1.12–2.74) | 1.69 (0.91–3.12) |

*CI* confidence interval, *PRS* polygenic risk score, *Q1* quartile 1, *Q3* quartile 3, *SD* standard deviation.
*Hazard ratios obtained using Cox proportional hazards models adjusted for age, sex, and principal components 1–5.
[†]N includes all individuals without the prevalent condition at baseline.
[‡]Includes n = 20 events with high confidence loss-of-function, deleterious missense, known pathogenic or likely pathogenic variant for HCM, and n = 50 events with high confidence loss-of-function, deleterious missense, known pathogenic or likely pathogenic rare variant for DCM (see text and Supplementary Table 18).

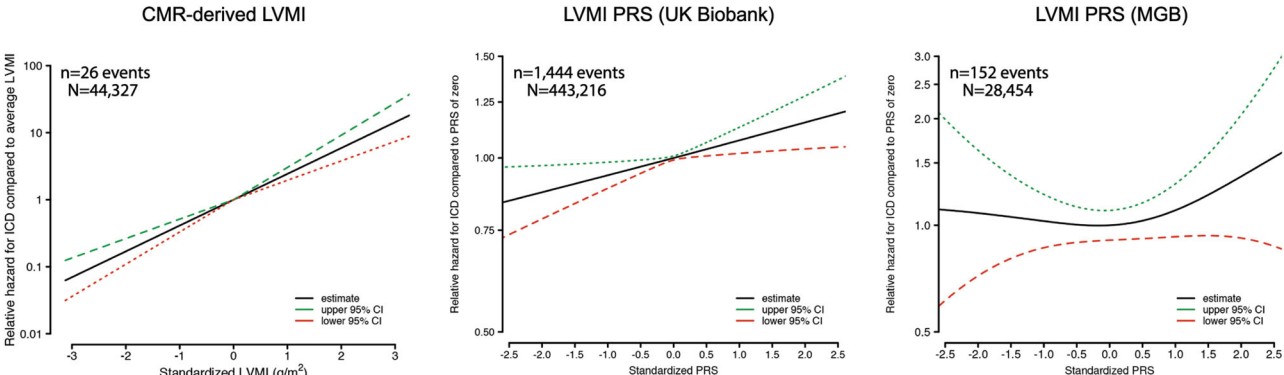

**Fig. 5 | Association between CMR-derived and genetically predicted LVMI and incident ICD implant.** Depicted are plots showing the relative hazard of incident implantable cardioverter-defibrillator (ICD) implant as a function of increasing standardized CMR-derived LVM index (left), increasing standardized LVMI PRS in UK Biobank (middle) and increasing standardized LVMI PRS in Mass General Brigham (MGB, right). In each plot, the y-axis depicts the relative hazard of incident ICD compared to the hazard observed for individuals with an average LVMI (left) or average PRS value (middle and right), derived from Cox proportional hazards models adjusted for age and sex (left), and adjusted for age, sex, and the first five principal components of genetic ancestry (middle and right). The relative hazard is plotted on the logarithmic scale. The functional form of the association was selected empirically using a penalized spline approach, in which the degrees of freedom for the penalized spline fit were chosen based on minimization of the corrected Akaike Information Criterion[75]. The number of events and individuals included in each analysis are listed above each plot.

genetic determinants of LVMI. Specifically, lead variant rs143800963 is located on chromosome 1 within 20 kb of *NPPA* and *NPPB*, genes that encode the natriuretic peptides Nppa and Nppb, respectively, with both proteins playing important roles in blood pressure regulation and salt homeostasis[26]. Both Nppa and Nppb are constitutively expressed in ventricular myocardium and upregulated in response to stress[27]. *NPPB* knockout in mice results in augmentation of the cardiac fibrosis response to pressure overload[28]. Conversely, cardiomyocyte-specific deletion of *ORAI1*, which encodes a regulator of calcium-induced calcium release, results in improved response to pressure overload and protection against angiotensin II-induced cardiac remodeling in adult myocardium[29]. *IGFR1*, an eQTL for LV tissue in which predicted expression in LV was associated with LVMI, encodes the insulin-like

growth factor receptor 1, which has been implicated in organ growth and insulin resistance[30].

Several LVMI candidate genes have previous links to cardiomyopathy and HF. The strongest association we observed was at rs2255167, a variant located in *TTN*, in which mutations have been previously associated with familial cardiomyopathy[31] and early-onset AF[32]. One of the loci detected in the European ancestry analysis (and suggestive in the primary analysis), *FLNC*, encodes filamin C, an actin-related protein associated with familial HCM[16], restrictive cardiomyopathy[17], arrhythmogenic cardiomyopathy[15], and LV contractile function[19]. A mouse knock-in of filamin C results in myofibrillar degeneration[33]. *PPP3CB*, which encodes the signaling protein calcineurin, has been implicated in pathologic cardiac hypertrophy[34]. Lead variant rs3729989 is located

near *MYBPC3*, a gene encoding the cardiac myosin-binding protein. Mutations in *MYBPC3* are a known cause of DCM and HCM[35, 36]. *FTO*, an obesity gene previously associated with HF[13], was associated with unindexed LV mass, but not LVMI. Interestingly, we identified several loci which are novel for LVM but have prior associations with electrocardiographic traits[37,38]. Future work is warranted to assess whether such associations may reflect electrical manifestations of LV mass or the presence of a cardiomyopathy.

Importantly, we observed that both phenotypic and genetically predicted LVMI were associated with increased risks of incident cardiovascular events. Increased LVMI and LVH are consistently associated with HF[2]. Here, we observed associations not only with HF, but also incident DCM, HCM, and insertion of an ICD (a surrogate for cardiomyopathy or ventricular arrhythmias). Consistent with the notion that LVMI may be an endophenotype for certain cardiomyopathies, we observed that genetically predicted LVMI (using a 465-variant PRS) was associated with greater risk of incident ICD implant in a separate set of UK Biobank participants as well as an external sample from the MGB healthcare system. Of note, we did not exclude individuals with DCM or HCM from our incident disease analyses since we hypothesized that polygenic risk may nevertheless contribute to the development of clinical outcomes[39]. In the context of low event rates, however, the LVMI PRS was associated with incident DCM only in the UK Biobank, and associations with incident HCM were not significant in either sample. Consistent with expectations[40, 41], using Mendelian-randomization analyses, we observed associations between genetically predicted blood pressure and diabetes risk with greater LVM. Overall, our findings provide evidence that the genetic variation underlying increased LVM may be clinically relevant, and highlight the need for future research to evaluate the potential utility of a polygenic predictor of LVM to improve identification of individuals at risk of incident cardiomyopathy.

Our study has limitations. First, our analysis was a mixed-ancestry GWAS, but the sample is predominantly of European descent. Therefore, our results may not generalize to individuals of other ancestries. Second, we used a previously published deep learning model (ML4H$_{seg}$) to facilitate well-powered GWAS of CMR-derived LVM. ML4H$_{seg}$ was trained using an imperfect segmentation method as ground truth[11, 42], which may have led to lower agreement with true LVM as compared to some alternative approaches (e.g., 95% limits of agreement −27g to 27 g with ML4H$_{seg}$ versus −18 to 18 g by Bai et al. using a proprietary deep learning model[43] and −5 to 8 g by Peterson et al. in a small set of hand-labeled measurements[44]). Nevertheless, estimates from ML4H$_{seg}$ correlate strongly ($r = 0.86$) with hand-labeled CMR-derived LVM in the UK Biobank[11], and MR analyses recapitulated a known causal relationship between elevated blood pressure and increased indexed LVM[40]. Third, our ability to assess for associations between CMR-derived LVMI and incident outcomes was limited by event rates and follow-up currently available after imaging. Fourth, generalizability may be affected by bias introduced by methods of enrollment, as UK Biobank participants are enriched for health and socioeconomic status compared to the general population[45]. Fifth, we analyzed LVM indexed to body surface area since this measure is in common clinical use, even though alternative methods of body mass correction exist. We therefore performed multiple analyses using alternative indexing methods (e.g., 2.7th power of height).

In summary, we performed GWAS of deep-learned CMR-derived LVM including nearly 50,000 individuals. We discovered 12 independent loci meeting genome-wide significance, including 11 that are novel. Using complementary downstream analyses, we identified multiple candidate genes, many of which are involved in cardiac structure and function, and several that have been previously implicated in cardiomyopathy. Both CMR-derived and genetically determined LVM were associated with incident ICD implant in independent datasets. Our findings add to our understanding of common genetic variation underlying LVM and demonstrate the potential to use deep learning to define rich phenotypes at scale to empower clinically relevant biological discovery.

## Methods

### Study populations
The discovery sample comprised the UK Biobank, a population-based prospective cohort of 502,629 participants recruited between 2006–2010 in the United Kingdom to investigate the genetic and lifestyle determinants of disease. The design of the cohort has been described previously[46, 47]. Briefly, approximately 9.2 million individuals aged 40-69 years living within 25 miles of the 22 assessment centers in England, Wales, and Scotland were invited, and 5.4% participated in the baseline assessment. Extensive questionnaire data, physical measures, and biological data were collected at recruitment, with ongoing data collection in large subsets of the cohort, including repeated assessments and multimodal imaging. At the time of the current analysis, over 450,000 individuals have genome-wide genotyping data available. All participants are followed up for health outcomes through linkage to national health-related datasets.

We utilized the MGB Biobank to replicate a LVMI PRS that we derived in the UK Biobank. The MGB Biobank is a biorepository comprising patients from a multi-institutional healthcare network spanning seven hospitals in the New England region of the United States. MGB Biobank participants are followed for health outcomes through linkage to electronic health record (EHR) data.

UK Biobank and MGB Biobank participants provided written informed consent. The UK Biobank was approved by the UK Biobank Research Ethics Committee (reference number 11/NW/0382) and the MGB Biobank by the MGB Institutional Review Board. Use of UK Biobank (application #17488) and MGB Biobank data were approved by the local MGB Institutional Review Board.

### Cardiac magnetic resonance acquisition
For all analyses, we included individuals who underwent CMR during a UK Biobank imaging assessment and whose bulk CMR data were available for download as of 04-01-2020 (Fig. 1). The full CMR protocol of the UK Biobank has been described in detail previously[48]. Briefly, all CMR examinations were performed in the United Kingdom on a clinical wide-bore 1.5 Tesla scanner (MAGNETOM Aera, Syngo Platform VD13A, Siemens Healthineers, Erlangen, Germany). All acquisitions used balanced steady-state free precession with typical parameters.

### Left ventricular mass estimation
We obtained CMR-derived LVM from all individuals with available CMR imaging using ML4H$_{seg}$[11]. ML4H$_{seg}$ is a convolutional neural network which identifies pixels corresponding to LV myocardium, which are then summed to estimate LV area and multiplied by slice thickness to estimate LV myocardial volume. LV myocardial volume is then multiplied by myocardial density ($1.05 \, g/cm^3$) to yield LVM. LVM estimates were calibrated to the sex-specific sample means using manually labeled LVM measurements which were available within a subset of the UK Biobank sample ($n = 4910$), where sex was classified using self-reported data. LVM estimates obtained using the described method have been shown to have very good correlation (Pearson $r$ 0.86) and agreement (mean absolute error 10 g) against manually labeled LVM in the UK Biobank[11]. LVM estimates were indexed for body surface area using the DuBois formula to yield LVMI[49]. A total of 59 (0.1%) individuals with outlying estimated LVM values (defined as falling outside 5 interquartile ranges from the median, or any value $\leq 0 \, g/m^2$ following calibration) were removed prior to analyses (Fig. 1). The distribution of CMR-derived LVM is shown in Supplementary Fig. 7.

### Genome-wide association study
To identify common genetic variation associated with CMR-derived LVM, we performed a GWAS of indexed LVM using BOLT-LMM v2.3.4[50],

which accounts for ancestral heterogeneity, cryptic population structure, and sample relatedness by fitting a linear mixed model with a Bayesian mixture prior as a random effect[19, 51, 52]. Previous evidence supports the use of LMM approaches to perform GWAS of admixed populations, which may provide favorable statistical power[51, 53, 54], and similar approaches have been taken previously[19, 51, 52]. The GWAS was performed among 43,230 individuals having undergone CMR imaging, after exclusion of individuals without genetic data meeting standard quality control metrics (e.g., no evidence of sex chromosome aneuploidy, outliers in heterozygosity and missing rates). Imputed variants were retained if the imputation information metric was ≥0.3. All variants with minor allele frequency <1% were excluded from the final analyses. Our model was adjusted for age at CMR acquisition, sex, array platform, and first five principal components of genetic ancestry, where sex was classified on the basis of genetic sex. Associations were considered statistically significant at the standard genome-wide significance level ($p = 5 \times 10^{-8}$). Lead single nucleotide polymorphisms (SNPs) were grouped into independent loci based on distance (±500 kb), with conditional analyses performed to assess for independent signals within windows. Variants having suggestive (i.e., $p < 1 \times 10^{-6}$) but not genome-wide significant associations were similarly tabulated. Genetic inflation was assessed by calculating the genomic control factor λ, inspecting quantile-quantile plots, and calculating the linkage disequilibrium score (LDSC) regression intercept using LDSC v1.0.1[55]. Observed scale heritability ($h^2$) was estimated using the slope of LDSC regression. We assessed for independent signals within genome-wide significant loci by a) performing GWAS while conditioning on the imputed allele dosage of each lead SNP found in the primary GWAS (excluding insertion-deletion variants), and b) performing GWAS while conditioning on the top variant on chromosome 17 alone (rs6503451), to assess whether the additional variant located 914 kb apart on chromosome 17 (rs199502, $r^2 = 0.37$), was independent. The primary GWAS was performed among individuals of all genetic ancestries.

We performed several secondary GWAS analyses. First, we performed analogous GWAS restricted to individuals of European genetic ancestry ($n = 39,187$). Second, we performed GWAS of unindexed LV mass (with and without adjustment for height and weight), as well as LV mass alternatively indexed using the 2.7th power of height[56]. Third, we performed a GWAS of LVMI after rank-based inverse normal transformation. Fourth, we performed GWAS of LVMI excluding individuals with prevalent myocardial infarction and heart failure.

## Bioinformatics and in silico functional analyses

We assessed whether genes within 500 kb of lead SNPs were related to cardiac gene expression using GTEx[57] version 8 cis-eQTL tissue data (dbGaP Study Accession phs000424.v8.p2). To maximize power to detect potential candidate genes, we considered eQTLs for both atrial appendage (AA) and LV tissue data[19, 58]. We included lead variants as well as strong proxy variants ($r^2 \geq 0.8$). We also quantified tissue-specific expression levels from bulk RNA sequencing data from GTEx[57] version 8 (dbGaP Study Accession phs000424.v8.p2). We evaluated the effects of predicted gene expression levels on LVMI by performing a transcriptome-wide association study (TWAS) using S-PrediXcan[59]. GTEx genotypes and normalized expression data in AA and LV tissues provided in the software were used as training sets to develop the prediction models. Prediction models between each gene-tissue pair were developed using elastic net regression. In total, we tested 6636 and 6008 associations in AA and LV, respectively. The significance threshold for S-PrediXcan was therefore set at $p = 0.05/(6636 + 6008)$, or $3.95 \times 10^{-6}$. We assessed for potential long-range chromatin interactions using Hi-C analysis in adult heart tissues obtained from the Myocardial Applied Genomics Network (MAGNet, www.med.upenn.edu/magnet) at the University of Pennsylvania[60].

We prioritized candidate genes on the basis of closest proximity to the lead variant, eQTLs, TWAS, tissue-specific expression levels, Hi-C

analysis, and biologic plausibility based on previously reported data. All prioritized genes were supported by at least two lines of evidence.

## Comparison to prior associations with LV measurements and cardiovascular traits

To assess whether the variants we identified in association with LVMI have been previously associated with other LV measurements, we compared our loci to those reported to have genome-wide associations with other LV measurements in prior analyses by Pirruccello et al.[19] and Aung et al.[10]. We performed an analogous search for associations with any cardiovascular disease or risk factor using the National Human Genome Research Institute GWAS Catalog[61]. For these analyses, we tabulated all associations including the same variant, a variant serving as a strong proxy ($r^2 \geq 0.80$), or a variant mapping to the same candidate gene.

## Polygenic risk score development

To develop a PRS as a genetic instrument for CMR-derived LVMI, we applied a pruning and thresholding approach to our LVMI GWAS results. After removing insertion-deletion variants and strand ambiguous (i.e., A/T and C/G) variants to facilitate replication, we developed and tested four separate candidate PRS utilizing each combination of two thresholds used to define index SNPs ($p = 1 \times 10^{-6}$ and $p = 1 \times 10^{-4}$) and two thresholds used to prune proxy SNPs ($r^2 = 0.3$ and $r^2 = 0.5$). We then selected the PRS explaining the greatest variance in LVMI within the derivation set, which ultimately comprised a set of 465 variants ($r^2 = 0.3$, $p = 1 \times 10^{-4}$, variance of LVMI explained = 0.084; +3.56 g/m² increase in LVMI per 1-standard deviation [1-SD] increase in PRS, $p < 0.01$).

## Outcomes association testing

We assessed for associations between CMR-derived LVMI and incident AF, myocardial infarction, HF, ventricular arrhythmias, DCM, HCM, and implantable cardioverter-defibrillator (ICD) within participants with follow-up clinical data available after the imaging visit. We assessed for analogous associations using LVH, which was defined as LVMI > 72 g/m² in men and >55 g/m² in women[44], and alternatively as the sex-specific 90th percentile of LVM[1]. Diseases were defined using combinations of self-report and inpatient International Classification of Diseases, 9th and 10th revision codes (Supplementary Data 1). Start of follow-up was defined at the time of CMR acquisition and spanned until the earliest of an incident event, death, or last follow-up. The date of last follow-up was dependent upon the availability of linked hospital data, and was therefore defined as March 31, 2021 for participants enrolled in England (93.6%) and Scotland (6.1%), and February 28, 2018 for participants enrolled in Wales (0.3%).

We performed analogous association testing between the LVMI PRS and the same set of incident cardiovascular events among individuals in the UK Biobank that did not undergo CMR ($n = 443,326$). Outcome and person-time definitions were similar, although start of follow-up was defined as the date of UK Biobank enrollment and blood sample collection. We also repeated association testing between the LVMI PRS and incident events in the independent MGB Biobank sample, using analogous models with person-time beginning at the date of blood sample collection and ending at an event, death, or last encounter in the electronic health record.

## Mendelian-randomization analyses of blood pressure and diabetes

As a form of validation of our LVM estimation, we sought to identify evidence of known causal associations between elevated blood pressure and increased LVM[40]. We therefore conducted two-sample Mendelian-randomization (MR) within individuals of genetic European ancestry in the UK Biobank sample. Given strong epidemiologic associations between diabetes and LVM[62], we performed analogous

MR analyses for diabetes. Genetic instruments for systolic blood pressure (SBP) and diastolic blood pressure (DBP) were derived from a recent GWAS[63]. The same set of SNPs was used for both systolic and diastolic blood pressure, but weights specific to systolic versus diastolic blood pressure were used for the systolic and diastolic Mendelian-randomization analysis, respectively[63]. Utilizing an 865 SNP instrument for SBP and DBP, we prioritized inverse-variance weighted (IVW) meta-analyses of the effect of each SNP on CMR-derived LVMI (and LVM) divided by the effect of the same SNP on SBP and DBP, respectively. We performed an analogous procedure using a 337 SNP instrument for diabetes[64]. Linear regression models were adjusted for age, sex, genotyping array, and the first ten principal components of genetic ancestry, to determine the beta coefficients and standard errors for the association of each SNP with the outcome (CMR-derived LVMI). These SNP-specific estimates were combined to conduct two-sample Mendelian randomization using the 'MendelianRandomization' package in R. Weighted median and MR-Egger analyses were performed secondarily to address potential invalid instruments and directional pleiotropy.

## Statistical analysis

We tested associations between CMR-derived LVM and incident AF, myocardial infarction, HF, ventricular arrhythmias, DCM, HCM, and ICD using Cox proportional hazards regression with adjustment for sex and age at CMR acquisition. We fit analogous models using LVH (defined using the thresholds described above) and the LVMI PRS as the primary exposures. Models including the PRS were additionally adjusted for the first five principal components of genetic ancestry. For the PRS outcomes analyses, we did not exclude individuals with pathogenic or likely pathogenic variants for HCM or DCM for the following reasons: (a) a substantial proportion of individuals with clinically confirmed HCM and DCM have no causal variant identified[14,65], (b) recent evidence suggests that polygenic background may play an important role in disease development even among individuals carrying mutations[39], and (c) rare variant information is not available in all individuals in our UKBB or MGB replication samples. To assess the frequency of pathologic rare variants among individuals with incident HCM and DCM events, we did tabulate carrier status of high confidence loss of function, deleterious missense, and known pathogenic or likely pathogenic variants in HCM and DCM genes as cataloged in ClinVar as of 2/9/2021. We also included high confidence loss-of-function variants using LOFTEE[66], a plug-in of VEP[67], and deleterious missense variants[68] using 30 in silico prediction tools presented in v4.1a of the dbnsfp database[69]. A full list of variants is shown in Supplementary Table 17.

Validity of the proportionality assumption was assessed using the Grambsch-Therneau test of correlation[70] as well as visual inspection of smoothed fits to Schoenfeld residuals versus time. Where present, substantial deviations from proportional hazards (observed only for age, sex, and certain principal components of ancestry), were modeled by including interaction terms with strata of person-time.

Statistical analyses were performed using R v4.0 (packages 'data.table' v1.13.6, 'ggplot2' v3.3.3,'survival' v3.2-7,'prodlim' v2019.11.13, 'MendelianRandomization' v0.5.0)[71, 72]. Except where otherwise noted, all two-tailed p-values <0.05 were considered statistically significant.

## Reporting summary

Further information on research design is available in the Nature Portfolio Reporting Summary linked to this article.

## Data availability

UK Biobank data are publicly available by application (https://www.ukbiobank.ac.uk/enable-your-research/register). LV mass estimates used for the current analysis are accessible to UK Biobank researchers as returned data (return ID #3290). The GWAS summary statistics generated in this study have been deposited in the Human Genome Research Institute GWAS Catalog[61] under accession codes GCST90244710 for LVMI (ftp://ftp.ebi.ac.uk/pub/databases/gwas/summary_statistics/GCST90244001-GCST90245000/GCST90244710/) and GCST0244711 for unindexed LVM (ftp://ftp.ebi.ac.uk/pub/databases/gwas/summary_statistics/GCST90244001-GCST90245000/GCST90244711/) and from the Downloads page of the Cardiovascular Disease Knowledge Portal (broadcvdi.org). The LVMI PRS developed in this study has been deposited to the Polygenic Score (PGS) Catalog[73] under accession code PGS003427 (https://www.pgscatalog.org/score/PGS003427/). Mass General Brigham (MGB) data contain identifiable protected health information and participants have not consented to data sharing; therefore, the data cannot be shared publicly or with controlled access. This research has been conducted using the UK Biobank Resource under Application #17488.

## Code availability

Data processing scripts used to perform the analyses described herein are available at https://github.com/shaankhurshid/lvmass_gwas[74].

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

## Acknowledgements

J.P.P. is supported by a John S. LaDue Memorial Fellowship. L.-C.W. is supported by NIH 1R01HL139731. S.H.C. is supported by the NIH NHLBI BioData Catalyst Fellows program. J.E.H. is supported by NIH (R01HL134893, R01HL140224, K24HL153669). S.A.L. is supported by NIH 1R01HL139731 and American Heart Association 18SFRN34250007. P.T.E. is supported by NIH 1R01HL092577, R01HL128914, K24HL105780, American Heart Association 18SFRN34110082, and Foundation Leducq 14CVD01. V.N. is supported by NIH T32HL007604.

## Author contributions

Conceptualization: S.K. and S.A.L.; Methodology: S.K., J.L., J.P.P., L.C.W., S.H.C., A.W.H., X.W., S.F.F., V.N., K.J.B., K.G.A., P.B., and A.A.P.; Supervision: P.T.E. and S.A.L.; Writing – original draft: S.K. and J.L.; Writing – review and editing: J.E.H., P.T.E., and S.A.L.

## Competing interests

J.P.P. has consulted for Maze Therapeutics. S.F.F. receives research support from Bayer AG and IBM. L.-C.W. receives research support from IBM to the Broad Institute. P.B. received research support from Bayer AG and IBM, and consults for Novartis. J.E.H. has received research support from Bayer AG and Gilead Sciences, has received research supplies from EcoNugenics, and is an employee of Flagship Pioneering as of January 2023. A.A.P. receives research support from Bayer AG, IBM, Intel, and Verily, and has consulted for Novartis and Rakuten. P.T.E. receives research support from Bayer AG, and has consulted for Bayer AG, Novartis, MyoKardia and Quest Diagnostics. S.A.L. has received research support from Bristol Myers Squibb/Pfizer, Bayer AG, Boehringer Ingelheim, and Fitbit, has consulted for Bristol Myers Squibb/Pfizer and Bayer AG, participated in research collaborations with IBM, and is an employee of Novartis Institute for Biomedical Research as of July 2022. Remaining authors declare no competing interests.
