## [Peer Review File · Nature Communications]

Clinical and Genetic Associations of Deep Learning-Derived Cardiac Magnetic Resonance-Based Left Ventricular MassREVIEWER COMMENTS

Reviewer #1 (Remarks to the Author):

Key results: This paper reports results of a GWAS on left ventricular mass indexed to body surface area (LVMI), in participants of the UK Biobank cardiac imaging extension. The LVMI was derived from a machine learning algorithm. 12 independent associations (11 novel locations) met significance, including genes previously associated with cardiac contractility and cardiomyopathy. A polygenic risk score analysis was verified in a separate UK Biobank cohort and in an external dataset.

Validity: The deep learning method used to compute LVM is described in reference 11, and is a U-Net trained on automatically identified contours obtained from the Siemens InlineVF algorithm. This is a pre-deep-learning method with known inaccuracies (Suinesiaputra et al, *Int J Cardiovasc Imaging* 2018;34:281–291) and is known to be inferior to many deep learning algorithms developed subsequently (see eg. Bai et al *JCMR* 2018;20:65 which was trained on manual contours available from UK Biobank repository, and the review by Chen et al *Front Cardiovasc Med* 2020;7:25). There appears to have been no quality control for InlineVF contours (as recommended by Suinesiaputra et al) in the training of the UNet. The limits of agreement in ref 11 against manual contours (Fig 4) are +/- 27 g, after linear bias correction, which is high compared with Bai et al (~18g). It is likely that a more accurate LVM method would result in higher sensitivity.

Significance: UK Biobank GWAS, PRS and MR analyses of CMR mass in UK Biobank have been and continue to be reported as the amount of data available increases (eg Aung et al). Machine learning has previously been used to get phenotypes at scale [Bai et al *Nature Medicine* 2020;26:1654–1662]. More significant loci were found in this study than Aung et al, as expected with a larger sample size. A full analysis of UK Biobank CMR studies will only be possible when the recruitment target of 100,000 cases is achieved. The machine learning method used does not appear to be state of the art since it was trained on an outdated Siemens algorithm with known quality issues. In fact it is not clear what is the advantage of the machine learning in this case since the InlineVF results are already available for the entire cohort (as used in Pirruccello et al *Nature Comms* 2020;11:2254) and the machine learning trained on these results essentially reproduce them (with some improvement as shown in reference 11, but still high variance).

Data and Methodology: There does not appear to be a quality check on phenotypes (see suggested improvements)

Analytical approach: I can't comment on the GWAS PRS or Mendelian randomization methods.

Suggested improvements:

1. Use exact numbers in the abstract: these would be from Table 1 for GWAS, PRS and the Mass General Brigham dataset. However 32,328 is stated for UK Biobank GWAS in the supplemental methods, but 43,271 in Table 1? The numbers should be consistent.
2. A more detailed quality control method should be used to exclude outliers in LVM. Negative mass estimates were removed but it is hard to see how that could occur. Some mass values in the density plot appear too low for adults. An inter-quartile range method (eg $5 \times \text{IQR}$) would be more robust to outliers.
3. The distribution of LVMI appears skewed and could be normalized as in Aung et al.
4. The unindexed LVM GWAS did not include height or weight as covariates which would control somewhat for body habitus. This should be run as a secondary analysis.
5. Indexing is a ratio method, known to have problems as a method for controlling for a covariate (Kronmal JR Stat Soc 1993;156:379-392). Indexing LVM to BSA has known problems (Whalley et al J Hypertension 1999; 17:569-74). Lean body mass could be available in UK Biobank and would be a better correction for body habitus. This should be added in the limitations section.
6. Individuals with a history of cardiovascular disease were not excluded from the GWAS (as done by Aung et al). This could obscure genetic relationships with LVM according to severity of disease. A secondary analysis without CVD would be interesting and a PRS could still be related to disease, which would have the benefit of not being confounded by disease present in the PRS definition.
7. Consider doing a Hi-C analysis for long range chromatin interactions.
8. Are the two loci which drop out of European analysis still borderline?
9. I could not see the number of individuals present in the analysis restricted to European ancestry? This should be very similar to the full GWAS since there were only small numbers recruited of other ethnicities in UK Biobank? What does this add?

Clarity and context: the manuscript is generally clear and well written.

References:

1. Reference 6 is odd for demonstrating the accuracy of CMR LVM: better would be Grothues et al Am Heart J. 2004;147(2):218-23

My expertise: I am able to review the machine learning and cardiac image analysis aspects. I am not an expert on GWAS or bioinformatics so was not able to assess this in detail.

Reviewer #2 (Remarks to the Author):

This is an excellent paper taking advantage of the UK Biobank as a resource. The manuscript focuses on LV mass index as the main phenotype and identifies a number of novel SNPs in addition to detecting signals for previously identified variants. Subsequent expression analysis using GTEx data further annotates the signals. The methodology of measuring LVM through CMR is a strength.

The authors should consider the following questions:

1. The authors index LVM to BSA. It is well established that this method can 'normalize' LVM in obese individuals. Another commonly used method is indexing to $\text{height}^2.7$. The authors should explore the associations signals for robustness.
2. The authors preform a mendelian randomization for hypertension. Diabetes is another well established risk factor and might warrant further examination.
3. The outcomes analysis is interesting and potentially clinically important. However, this will require further descriptions. First, the selected outcomes are diverse. While AF and HF appear as expected progression of LVH, DCM or HCM could be understood from a genetic perspective as a more monogenic disease. Are the authors hereby suggesting that HCM can be also caused by a combination of variants from a PRS, in addition to the well established mutations in HCM genes? This needs further clarification. Same holds for DCM. Furthermore, are patients with known mutations in genes for DCM or HCM excluded and if not how could this affect the analysis? It could appear that even a small number of patients with mutations in known HCM genes could impact the results.

Furthermore, the authors describe a progression to HF. If a PRS could be suggested as a prognostic marker, this will also need further detail. Obviously, a previous reduction on LV function is likely to further accelerate the progression to HF, as its treatment. The authors might want to consider a more detailed analysis related to HF which includes LV function parameters at baseline as well as medication use to explore the role of the PRS. Without this information, the results are potentially interesting, however a potential clinical use might be less clear.

Reviewer #3 (Remarks to the Author):

In this study the authors conduct a genome-wide association study (GWAS) of left-ventricular mass (LVM) as predicted using a deep learning algorithm. The 12 loci highlighted by this approach appear to be biologically plausible and the application of a polygenic risk score using these findings is associated with risk of cardiomyopathies in independent samples.

For the most part, this study is appropriately conducted and the manuscript is well written. I have included several comments below to try and help refine this work. In particular, I found the eQTL/TWAS and MR components to be the weakest parts of the manuscript.

1. The main question I had regarding the GWAS results is how many novel associations have been identified by previous GWAS of heart structure phenotypes? For example, several authors on this work previously conducted a GWAS of MRI-derived left ventricular phenotypes in the UK Biobank subset of participants:

<https://www.nature.com/articles/s41467-020-15823-7>

This previous work is referenced in the current manuscript (ref 21) - although I found it strange that this is in relation to the source of eQTL data used rather than in relation to the loci highlighted by this new study.

I would ask that the authors provide a summary of how the 12 loci associated with LVM in this new study relate to the various heart structure phenotypes using results from the study linked above as all these summary statistics are available. A summary table would be helpful for this to compare side by side estimates. A quick skim looks like there will be overlapping GWAS loci (e.g. FLNC) - so this will help give the reader an idea about how many have been previously reported to play a role in heart structure phenotypic variation.

2. Did the authors take any further steps to account for the non-European subset of participants in their GWAS? How did they account for the allele frequency differences amongst participants? Most of the literature I am familiar with has conducted GWAS of participants of different ancestries separately before meta-analysing findings together. I think it would be good for the authors to justify this aspect of their study design with references to previous studies who have undertaken a similar approach. I appreciate that the European only GWAS results are available to the reader, but I think the decision to effectively ignore ancestry in the primary GWAS would be worthwhile justifying as some readers will likely be unfamiliar with this approach.

3. The eQTL/TWAS section seems like quite a light touch in terms of how much of the manuscript discusses this analysis. How well do the authors believe this strategy helped map GWAS signals to causal genes? For example, in supplementary table 2 the TTN locus has 'FKBP7' in the corresponding 4 columns relating to eQTL/TWAS and yet the authors prioritise TTN based on 'strong biologic plausibility'. Does this not diminish the utility of this SNP to gene mapping strategy i.e. just ignoring the findings? Can the authors comment on loci where they believe this approach was worthwhile?

4. I also think that the Mendelian randomization analysis is the weakest section of the paper. Findings from this analysis are not mentioned in the discussion at all. I also have no idea what the term 'genetically mediated' means. Are the authors able to clarify?

Overall, I don't think this section really adds much to the paper. I don't believe this is a compelling 'validation' of the LVM measure as the authors suggest and think that the PRS analysis is a lot stronger in this respect...I'm struggling to see the value added by the MR.

If the authors feel very strongly about retaining it then to start with I would recommend changing the interpretation of findings to 'genetically predicted' rather than 'mediated'. The instrument selection is also quite strange...the authors use blood pressure findings from a study which is over a decade old. Maybe they are concerned about sample overlap in the UKB cohort but I don't think this will be too detrimental in terms of overfitting given that LVM is based on a subset of UKB. Is there also not a more recent blood pressure GWAS since 2011 which didn't include UKB?

We thank the editors and reviewers for their comments, which we feel have strengthened the manuscript.

Reviewer #1:

R1.1. Key results: This paper reports results of a GWAS on left ventricular mass indexed to body surface area (LVMI), in participants of the UK Biobank cardiac imaging extension. The LVMI was derived from a machine learning algorithm. 12 independent associations (11 novel locations) met significance, including genes previously associated with cardiac contractility and cardiomyopathy. A polygenic risk score analysis was verified in a separate UK Biobank cohort and in an external dataset.

Author Response: We thank the reviewer for their thoughtful read and comments.

R1.2. Validity: The deep learning method used to compute LVM is described in reference 11, and is a U-Net trained on automatically identified contours obtained from the Siemens InlineVF algorithm. This is a pre-deep-learning method with known inaccuracies (Suinesiaputra et al, Int J Cardiovasc Imaging 2018;34:281–291) and is known to be inferior to many deep learning algorithms developed subsequently (see eg. Bai et al JCMR 2018;20:65 which was trained on manual contours available from UK Biobank repository, and the review by Chen et al Front Cardiovasc Med 2020;7:25). There appears to have been no quality control for InlineVF contours (as recommended by Suinesiaputra et al) in the training of the UNet. The limits of agreement in ref 11 against manual contours (Fig 4) are +/- 27 g, after linear bias correction, which is high compared with Bai et al (~18g). It is likely that a more accurate LVM method would result in higher sensitivity.

Author Response: We agree that the deep learning model we used to estimate LV mass is imperfect, and that error in LV mass estimation may reduce our ability to detect variants associated with CMR-derived LV mass. Nevertheless, we note that the ML4H_{seg} model we utilized offers favorable agreement with hand-labeled contours as compared to inlineVF.¹ In our prior study, we observed correction of gross segmentation errors using ML4H_{seg} as opposed to inlineVF, suggesting that improved accuracy may be related to a lower likelihood of committing gross segmentations errors using ML4H_{seg}.¹ Consistent with this interpretation, ML4H_{seg} continued to outperform inlineVF even after applying the linear bias correction suggested by Suinesiaputra et al.¹

We note that the alternative deep learning models referenced by the reviewer and their corresponding trained weights are not publicly available. In response to the reviewer's comment we acknowledge limitations in model accuracy in the Discussion. We additionally respond to specific quality control suggestions made by this reviewer below.

Revised manuscript

Discussion (Page 22, Line 21): "Second, we used a deep learning model to estimate CMR-derived LVM, and therefore imperfect accuracy of estimations may have lead to reduced functional power to detect genetic associations. Nevertheless, we note that the model we utilized (ML4H_{seg}) is accurate, having a correlation of r=0.86 with hand-labeled CMR-derived LVM in the UK Biobank,² and MR analyses recapitulated a known causal relationship between elevated blood pressure and increased indexed LVM.^{3"}

R1.3. Significance: UK Biobank GWAS, PRS and MR analyses of CMR mass in UK Biobank have been and continue to be reported as the amount of data available increases (eg Aung et al). Machine learning has previously been used to get phenotypes at scale [Bai et al Nature Medicine 2020;26:1654–1662]. More significant loci were found in this study than Aung et al, as expected with a larger sample size. A full analysis of UK Biobank CMR studies will only be possible when the recruitment target of 100,000 cases is achieved. The machine learning method used does not appear to be state of the art since it was trained on an outdated Siemens algorithm with known quality issues. In fact it is not clear what is the advantage of the machine learning in this case since the InlineVF results are already available for the entire cohort (as used in Pirruccello et al Nature Comms 2020;11:2254) and the machine learning trained on these results essentially reproduce them (with some improvement as shown in reference 11, but still high variance).

Author Response: As discussed in response to this reviewer's previous comment, we agree that our deep learning model has limitations. However, we submit that even modest increases in accuracy (e.g., as provided by ML4H_{seg} as compared to inlineVF, with and without linear bias correction¹) may optimize power to detect true genetic associations. We respond to specific suggestions made by this reviewer below.

R1.4. Data and Methodology: There does not appear to be a quality check on phenotypes (see suggested improvements)

Author Response: In response to this reviewer's suggestion we have modified our quality control steps for the LVMI phenotype, as detailed in response to the reviewer's specific comments below.

R1.5. Use exact numbers in the abstract: these would be from Table 1 for GWAS, PRS and the Mass General Brigham dataset. However 32,328 is stated for UK Biobank GWAS in the supplemental methods, but 43,271 in Table 1? The numbers should be consistent.

Author Response: We thank the reviewer for this suggestion. In response, we now report exact numbers in the Abstract and have corrected any inconsistencies. As described below, in response to reviewer comments resulting in changes in our quality control, as well as removal of a few individuals for revoked consent, the final GWAS sample size has changed slightly to N=43,230.

Revised manuscript:

Abstract: "In the current study, we performed a genome-wide association study (GWAS) of CMR-derived LVM indexed to body surface area (LVMI) estimated using a deep learning algorithm within 43,230 participants from the UK Biobank."

R1.6. A more detailed quality control method should be used to exclude outliers in LVM. Negative mass estimates were removed but it is hard to see how that could occur. Some mass values in the density plot appear too low for adults. An inter-quartile range method (eg 5*IQR) would be more robust to outliers.

Author Response: In response to this reviewer's suggestion we have modified our quality control steps for the LVMI phenotype and explain in greater detail. We now adopt the reviewer's suggested method of outlier removal, eliminating individuals with estimated LVMI beyond the median ± 5 interquartile ranges. We also continue to remove values $\leq 0g$, which result from linear recalibration, as described previously with ML4H_{seg}.¹

With this change, in addition to several additional individuals excluded for revoked consent, the total sample size with available LV mass measurements has decreased from 44,418 previously to 44,375 currently. The distributions of LVM and LVMI after outlier removal are shown in **Supplementary Figure 1**.

Revised manuscript:

Methods (Page 7, Line 18): “A total of 59 (0.1%) individuals with outlying estimated LVM values (defined as falling outside 5 interquartile ranges from the median, or any value ≤ 0 g following recalibration) were removed prior to analyses (**Figure 1**). The distribution of CMR-derived LVM is shown in **Supplementary Figure 1**.

Supplementary Figure 1. Distribution of CMR-derived LV mass

Depicted is the distribution of CMR-derived LV mass (LVM, left) and LV mass index (LVMI, right) within the UK Biobank phenotypic sample (N=44,375). The y-axis depicts the relative probability of an encountering a given value on the x-axis. Four high outlying observations for LVM are not shown for graphical purposes.

R1.7. The distribution of LVMI appears skewed and could be normalized as in Aung et al.

Author Response: We submit that deviations from normality in the LVMI distribution are relatively minor, and therefore that in the context of our sample size a GWAS of untransformed LVMI is valid. Furthermore, our untransformed GWAS has the additional advantage that the allele effect sizes are interpretable as average effects on LVMI in g/m². Nevertheless, in response to the reviewer’s comment, we now include a secondary analysis in which we perform a GWAS of LVMI after rank-based inverse normal transformation. The results of this analysis are similar to the untransformed LVMI GWAS and are shown in **Supplementary Table 8**. Specifically, of the 11 independent associations identified in the primary GWAS, 10 are significant in the rank-based inverse normal transformed analysis, and the remaining association (rs62621197 near ADAMTS10) meets the suggestive threshold ($p=6.6 \times 10^{-8}$). All effect sizes have consistent directions. In response to additional reviewer comments, we have also performed several other new secondary analyses, the results of which are also summarized in **Supplementary Table 12**.

Revised manuscript:

Methods (Page 8, Line 22): “We performed several secondary GWAS analyses. First, we performed analogous GWAS restricted to individuals of European genetic ancestry (n=39,187). Second, we performed GWAS of unindexed LV mass (with and without adjustment for height and weight), as well as LV mass alternatively indexed using the 2.7th power of height.⁴ Third, we performed a GWAS of LVMI after rank inverse normal transformation. Fourth, we performed GWAS of LVMI excluding individuals with prevalent myocardial infarction and heart failure. Further details of GWAS methods are provided in the **Supplemental Methods.**”

Results (Page 15, Line 18): “Results of secondary GWAS analyses, including rank-based inverse normal transformed LVMI, LVMI indexed using the 2.7th power of height, LVMI indexed using lean body mass, LVMI with exclusions for prevalent myocardial infarction and heart failure, and unindexed LVM adjusted for height and weight, are shown in **Tables 7-11**. Results obtained using alternative indexing methods were broadly consistent with the primary analysis, both in terms of variants identified and effect directions. A summary of association results for the lead variants identified in the primary GWAS tested across varying indexing methods is shown in **Table 12.**”

Supplementary Table 8. Variants associated with CMR-derived left ventricular mass after rank-based inverse normal transformation

rsID	Chr	Position	Closest gene(s)	Function	Risk/alt allele	RAF	Beta	SE	P-value
		(hg 38)							
rs143800963	1	11835418	CLCN6	Intronic	C/A	0.95	0.077	0.0133	6.5x10 ⁻⁹
rs2255167*	2	178693555	TTN	Intronic	T/A	0.81	0.079	0.0075	1.5x10 ⁻²⁶
rs10497529†	2	178975161	CCDC141	Missense	G/A	0.96	0.100	0.0165	6.9x10 ⁻¹⁰
rs36034102	4	80280894	FGF5	Intronic	G/T	0.73	-0.038	0.0068	2.7x10 ⁻⁸
-	5	133066736	HSPA4	Indel	CTT/C	0.72	0.041	0.0067	1.1x10 ⁻⁹
rs9388498	6	126552277	CENPW	-	G/T	0.81	-0.045	0.0079	3.4x10 ⁻⁹
rs76540339	10	18538957	CACNB2	Intronic	A/C	0.87	0.051	0.0087	8.2x10 ⁻⁹
rs34163229	10	73647154	SYNPO2L	Missense	G/T	0.86	-0.049	0.0085	6.6x10 ⁻⁹
rs3729989	11	47348490	MYBPC3	Missense	T/C	0.87	-0.051	0.0087	8.1x10 ⁻⁹
rs28552516	12	121592356	KDM2B	Intronic	C/T	0.85	-0.050	0.0083	2.5x10 ⁻⁹
rs6598541	15	98727906	IGF1R	Intronic	A/G	0.36	-0.035	0.0062	3.9x10 ⁻⁸
rs11376559	16	14995819	PDXDC1	Intronic	C/CA	0.70	0.039	0.0066	2.4x10 ⁻⁹
rs6503451	17	45870981	MAPT	Intronic	T/C	0.67	-0.043	0.0065	4.8x10 ⁻¹¹
rs199502‡	17	46785247	WNT3	Intronic	G/A	0.79	-0.044	0.0073	1.5x10 ⁻⁹

*Locus previously reported for LVM⁶

†Variant identified in conditional analysis conditioned on lead SNPs (beta, standard error, and p-value are adjusted)

‡Association no longer observed in analysis conditioned on rs6503451

Chr., chromosome; RAF, risk allele frequency; OR, odds ratio.

Table 12. Associations with CMR-derived left ventricular mass across varying indexing methods

				Body surface area		Body surface area (rank-inverse normal transformation)		2.7 th power of height		Lean body mass	
rsID*	Chr	Position (hg38)	Closest gene(s)	Beta [†]	P-value	Beta [†]	P-value	Beta [†]	P-value	Beta [†]	P-value
rs143800963	1	11835418	CLCN6	0.078	4.2x10 ⁻⁹	0.077	6.5x10 ⁻⁹	0.073	2.5x10 ⁻⁷	0.082	3.9x10 ⁻⁸
rs2255167	2	178693555	TTN	0.079	3.2x10 ⁻²⁶	0.079	1.5x10 ⁻²⁶	0.071	1.4x10 ⁻¹⁸	0.094	7.2x10 ⁻²⁹
rs10497529 [‡]	2	178975161	CCDC141	0.100	3.0x10 ⁻¹⁰	0.100	6.9x10 ⁻¹⁰	0.095	6.0x10 ⁻⁸	0.120	8.8x10 ⁻¹¹
-	5	133066736	HSPA4	0.041	1.6x10 ⁻⁹	0.041	1.1x10 ⁻⁹	0.040	2.7x10 ⁻⁸	0.049	4.0x10 ⁻¹¹
rs9388498	6	126552277	CENPW	-0.045	4.1x10 ⁻⁹	-0.045	3.4x10 ⁻⁹	-0.042	3.4x10 ⁻⁷	-0.045	2.1x10 ⁻⁷
rs34163229	10	73647154	SYNPO2L	-0.049	1.0x10 ⁻⁸	-0.049	6.6x10 ⁻⁹	-0.049	1.1x10 ⁻⁷	-0.051	9.8x10 ⁻⁸
rs3729989	11	47348490	MYBPC3	-0.050	1.8x10 ⁻⁸	-0.051	8.1x10 ⁻⁹	-0.047	4.9x10 ⁻⁷	-0.049	8.0x10 ⁻⁷
rs28552516	12	121592356	KDM2B	-0.047	1.5x10 ⁻⁸	-0.050	2.5x10 ⁻⁹	-0.052	4.0x10 ⁻⁹	-0.054	7.3x10 ⁻⁹
rs6598541	15	98727906	IGF1R	-0.034	4.6x10 ⁻⁸	-0.035	3.9x10 ⁻⁸	-0.035	1.8x10 ⁻⁷	-0.043	1.0x10 ⁻⁹
rs56252725	16	14995819	PDXDC1	0.044	3.7x10 ⁻⁹	0.044	3.0x10 ⁻⁹	0.039	9.0x10 ⁻⁷	0.045	8.5x10 ⁻⁸
rs6503451	17	45870981	MAPT	-0.042	1.1x10 ⁻¹⁰	-0.043	4.8x10 ⁻¹¹	-0.045	1.3x10 ⁻¹⁰	-0.041	1.9x10 ⁻⁸
rs199501	17	46785247	WNT3	0.043	1.1x10 ⁹	0.042	3.1x10 ⁻⁹	0.043	1.0x10 ⁻⁸	0.039	6.1x10 ⁻⁷
rs62621197	19	8605262	ADAMTS10	0.091	2.9x10 ⁻⁸	0.088	6.6x10 ⁻⁸	0.060	4.9x10 ⁻⁴	0.110	7.6x10 ⁻¹⁰

*Variants shown are those significant in the primary GWAS

†All betas are presented per 1-standard deviation increase to facilitate comparisons

‡Variant identified in conditional analysis conditioned on lead SNPs using respective indexing method (beta, standard error, and p-value are adjusted)

R1.8. The unindexed LVM GWAS did not include height or weight as covariates which would control somewhat for body habitus. This should be run as a secondary analysis.

Author Response: In response to the reviewer’s comment, we now include a secondary analysis in which we perform GWAS of unindexed LVM with height and weight included as covariates. The results of this analysis are shown in **Supplementary Table 11**.

Revised manuscript:

Methods (Page 8, Line 22): “We performed several secondary GWAS analyses. First, we performed analogous GWAS restricted to individuals of European genetic ancestry (n=39,187). Second, we performed GWAS of unindexed LV mass (with and without adjustment for height and weight), as well as LV mass alternatively indexed using the 2.7th power of height.⁴ Third, we performed a GWAS of LVMI after rank inverse normal transformation. Fourth, we performed GWAS of LVMI excluding individuals with prevalent myocardial infarction and heart failure. Further details of GWAS methods are provided in the **Supplemental Methods.**”

Results (Page 15, Line 18): “Results of secondary GWAS analyses, including rank-based inverse normal transformed LVMI, LVMI indexed using the 2.7th power of height, LVMI indexed using lean body mass, LVMI with exclusions for prevalent myocardial infarction and heart failure, and unindexed LVM adjusted for height and weight, are shown in **Supplementary Tables 7-11.**”

Table 11. Variants associated with unindexed CMR-derived left ventricular mass conditioned on height and weight

rsID	Chr	Position	Closest gene(s)	Function	Risk/alt allele	RAF	Beta	SE	P-value
		(hg 38)							
rs143800963	1	11835418	CLCN6	Intronic	C/A	0.95	1.64	0.297	3.30x10 ⁻⁸
rs2255167	2	178693555	TTN	Intronic	T/A	0.81	1.87	0.168	3.20x10 ⁻²⁹
-	3	169759203	ACTRT3	Indel	CAA/C	0.75	0.85	0.154	4.50x10 ⁻⁸
-	5	133066736	HSPA4	Indel	CTT/C	0.72	0.85	0.149	1.00x10 ⁻⁸
rs62388970	5	148528822	HTR4	Intronic	G/A	0.62	-0.77	0.137	1.70x10 ⁻⁸
rs13198983	6	126863841	RSPO3	-	G/A	0.60	-0.76	0.139	4.10x10 ⁻⁸
rs848466	7	77963290	PHTF2	-	T/C	0.49	-0.78	0.133	4.00x10 ⁻⁹
rs13230127	7	128859806	ATP6V1F	Intronic	C/T	0.87	1.13	0.196	1.10x10 ⁻⁸
rs35886223	7	129370466	AHCYL2	Intronic	G/A	0.67	0.80	0.144	2.40x10 ⁻⁸
rs34163229	10	73647154	SYNPO2L	Missense	G/T	0.86	-1.21	0.191	2.00x10 ⁻¹⁰
rs111555687	10	89819910	KIF20B	-	T/C	0.98	2.54	0.448	1.40x10 ⁻⁸
rs4980386	11	1874478	LSP1	Intronic	C/A	0.62	-0.83	0.137	1.50x10 ⁻⁹
rs35443	12	115115073	TBX3	-	G/C	0.61	0.74	0.136	4.20x10 ⁻⁸
rs6598541	15	98727906	IGF1R	Intronic	A/G	0.36	-0.81	0.139	5.50x10 ⁻⁹
rs11376559	16	15037797	PDXDC1	Indel	C/CA	0.70	0.83	0.146	1.20x10 ⁻⁸
rs77727624	17	45713404	CRHR1	Intronic	A/G	0.78	-1.02	0.161	1.60x10 ⁻¹⁰
rs62073222	17	46297344	LRRC37A	Synonymous	A/G	0.27	1.12	0.162	5.90x10 ⁻¹²
rs151269919	17	46816119	WNT3	Indel	A/ACACA...	0.36	0.80	0.144	3.80x10 ⁻⁸

*Locus previously reported for LVM⁵
Chr., chromosome; RAF, risk allele frequency

R1.9. Indexing is a ratio method, known to have problems as a method for controlling for a covariate (Kronmal JR Stat Soc 1993;156:379-392). Indexing LVM to BSA has known problems (Whalley et al J Hypertension 1999; 17:569-74). Lean body mass could be available in UK Biobank and would be a better correction for body habitus. This should be added in the limitations section.

Author Response: We chose to perform GWAS of LVM indexed using body surface area (BSA) because this method is in common clinical use. We now outline our rationale in the manuscript. Nevertheless, in response to the reviewer's comment, we now also include a secondary analysis in which we index LVM by lean body mass. Results of the latter analysis are shown in **Supplementary Table 9**. The results of multiple secondary GWAS are also summarized in **Supplementary Table 12** (reproduced in response to R1.7 above).

Revised manuscript:

Methods (Page 8, Line 22): "We performed several secondary GWAS analyses. First, we performed analogous GWAS restricted to individuals of European genetic ancestry (n=39,187). Second, we performed GWAS of unindexed LV

mass (with and without adjustment for height and weight), as well as LV mass alternatively indexed using the 2.7th power of height.⁴ Third, we performed a GWAS of LVMI after rank inverse normal transformation. Fourth, we performed GWAS of LVMI excluding individuals with prevalent myocardial infarction and heart failure. Further details of GWAS methods are provided in the **Supplemental Methods.**”

Results (Page 15, Line 18): “Results of secondary GWAS analyses, including rank-based inverse normal transformed LVMI, LVMI indexed using the 2.7th power of height, LVMI indexed using lean body mass, LVMI with exclusions for prevalent myocardial infarction and heart failure, and unindexed LVM adjusted for height and weight, are shown in **Supplementary Tables 7-11**. Results obtained using alternative indexing methods were broadly consistent with the primary analysis, both in terms of variants identified and effect directions. A summary of association results for the lead variants identified in the primary GWAS tested across varying indexing methods is shown in **Supplementary Table 12.**”

Discussion (Page 23, Line 9): “Fifth, we analyzed LVM indexed using body surface area since this measure is in common clinical use, even though alternative methods of body mass correction exist. We therefore performed multiple analyses of alternative indexing methods (e.g., lean body mass).”

Supplementary Table 9. Variants associated with CMR-derived left ventricular mass indexed using lean body mass

rsID	Chr	Position	Closest gene(s)	Function	Risk/alt allele	RAF	Beta	SE	P-value
		(hg 38)							
rs143800963	1	11835418	CLCN6	Intronic	C/A	0.95	0.08	0.0149	3.9x10 ⁻⁸
rs4233937	2	66525119	MEIS1	Intronic	A/G	0.39	0.04	0.0068	3.8x10 ⁻⁸
rs2255167*	2	178693555	TTN	Intronic	T/A	0.81	0.09	0.0084	7.2x10 ⁻²⁹
rs36034102	4	80280894	FGF5	Intronic	G/T	0.73	-0.04	0.0076	1.5x10 ⁻⁸
-	5	133066736	HSPA4	Indel	CTT/C	0.72	0.05	0.0075	4.0x10 ⁻¹¹
rs73238147	7	128829863	FLNC	Intronic	T/C	0.86	0.05	0.0097	3.8x10 ⁻⁸
rs72814544	10	70695167	ADAMTS14	Intronic	G/A	0.77	0.04	0.0079	3.2x10 ⁻⁸
rs28489288	12	121587314	KDM2B	Intronic	A/G	0.85	-0.05	0.0093	7.0x10 ⁻⁹
rs6598541	15	98727906	IGF1R	Intronic	A/G	0.36	-0.04	0.0070	8.6x10 ⁻¹⁰
rs5816114	16	15005317	PDXDC1	Indel	A/AAAAAAG	0.65	0.04	0.0079	2.4x10 ⁻⁸
rs62073222	17	46297344	LRRC37A	Synonymous	A/G	0.27	0.05	0.0082	1.3x10 ⁻⁸
rs62621197	19	8605262	ADAMTS10	Missense	C/T	0.96	0.11	0.0185	7.6x10 ⁻¹⁰

*Locus previously reported for LVM⁵
Chr., chromosome; RAF, risk allele frequency; OR, odds ratio.

R1.10. Individuals with a history of cardiovascular disease were not excluded from the GWAS (as done by Aung et al). This could obscure genetic relationships with LVM according to severity of disease. A secondary analysis without CVD would be interesting and a PRS could still be related to disease, which would have the benefit of not being confounded by disease present in the PRS definition.

Author Response: We agree with the reviewer’s comment, and in response we provide a secondary analysis in which we perform GWAS of LVMI while excluding individuals with prevalent myocardial infarction and heart failure (i.e., the exclusion criteria utilized by Aung et al.⁵ in the referenced study). The results of this GWAS are similar to those of the primary analysis and are shown in **Supplementary Table 10** (reproduced below). As suggested, we also developed a separate PRS using this GWAS. The PRS derived with exclusion of individuals with prevalent myocardial infarction and heart failure shows generally similar associations with incident disease as the original PRS (shown in new **Supplementary Table 14**, reproduced below).

Revised manuscript:

Methods (Page 9, Line 4): “Fourth, we performed GWAS of LVMI excluding individuals with prevalent myocardial infarction and heart failure.”

Results (Page 15, Line 18): “Results of secondary GWAS analyses, including rank-based inverse normal transformed LVMI, LVMI indexed using the 2.7th power of height, LVMI indexed using lean body mass, LVMI with exclusions for prevalent myocardial infarction and heart failure, and unindexed LVM adjusted for height and weight, are shown in **Supplementary Tables 7-11.**”

Results (Page 18, Line 13): “Disease association results were generally similar in analyses restricted to individuals of European ancestry (**Supplementary Table 14**), and when utilizing a PRS derived from GWAS performed after exclusion of individuals with prevalent myocardial infarction and heart failure (**Supplementary Table 15**).”

Table 10. Variants associated with CMR-derived indexed left ventricular mass excluding individuals with heart failure and coronary artery disease

rsID	Chr	Position	Closest gene(s)	Function	Risk/alt allele	RAF	Beta	SE	P-value
		(hg 38)							
rs143800963	1	11835418	CLCN6	Intronic	C/A	0.95	0.08	0.0149	3.9x10 ⁻⁸
rs4233937	2	66525119	MEIS1	Intronic	A/G	0.39	0.04	0.0068	3.8x10 ⁻⁸
rs2255167*	2	178693555	TTN	Intronic	T/A	0.81	0.09	0.0084	7.2x10 ⁻²⁹
`	2	178975161	CCDC141	Missense	G/A	0.96	0.12	0.0185	8.8x10 ⁻¹¹
rs36034102	4	80280894	FGF5	Intronic	G/T	0.73	-0.04	0.0076	1.5x10 ⁻⁸
-	5	133066736	HSPA4	Indel	CTT/C	0.72	0.05	0.0075	4.0x10 ⁻¹¹
rs73238147	7	128829863	FLNC	Intronic	T/C	0.86	0.05	0.0097	3.8x10 ⁻⁸
rs72814544	10	70695167	ADAMTS14	Intronic	G/A	0.77	0.04	0.0079	3.2x10 ⁻⁸
rs28489288	12	121587314	KDM2B	Intronic	A/G	0.85	-0.05	0.0093	7.0x10 ⁻⁹
rs6598541	15	98727906	IGF1R	Intronic	A/G	0.36	-0.04	0.0070	8.6x10 ⁻¹⁰
rs5816114	16	15005317	PDXDC1	Indel	A/AAAAAAG	0.65	0.04	0.0079	2.4x10 ⁻⁸
rs62073222	17	46297344	LRRC37A	Synonymous	A/G	0.27	0.05	0.0082	1.3x10 ⁻⁸
rs62621197	19	8605262	ADAMTS10	Missense	C/T	0.96	0.11	0.0185	7.6x10 ⁻¹⁰

*Locus previously reported for LVM⁵

†Variant identified in conditional analysis conditioned on lead SNPs (beta, standard error, and p-value are adjusted)

Chr., chromosome; RAF, risk allele frequency

Table 15. Associations between LVMI PRS and incident disease using PRS derived among individuals without prevalent myocardial infarction and heart failure

Disease	N events / N total [†]	Follow-up, yrs (Q1, Q3)	Hazard ratio for covariate (95% CI)*		
			PRS (per 1 SD)	PRS (90 th percentile)	PRS (95 th percentile)
UK Biobank					
Atrial fibrillation	25050 / 435917	11.8 (11.0, 12.6)	1.02 (1.01-1.03)	1.03 (0.99-1.07)	1.06 (1.00-1.12)
Myocardial infarction	13405 / 432044	11.8 (11.0, 12.6)	1.02 (1.00-1.04)	1.04 (0.98-1.10)	1.03 (0.96-1.12)
Heart failure	13540 / 440590	11.9 (11.0, 12.6)	1.03 (1.02-1.05)	1.08 (1.02-1.14)	1.13 (1.05-1.22)
Ventricular arrhythmias	4882 / 442295	11.9 (11.1, 12.6)	1.05 (1.02-1.08)	1.12 (1.02-1.22)	1.13 (1.00-1.28)
Dilated cardiomyopathy	1023 / 443013	11.9 (11.1, 12.6)	1.08 (1.02-1.15)	1.10 (0.90-1.34)	1.29 (1.00-1.65)
Hypertrophic cardiomyopathy	420 / 443150	11.9 (11.1, 12.6)	1.11 (1.00-1.22)	1.02 (0.74-1.40)	1.29 (0.87-0.92)
Implantable defibrillator	1444 / 443216	11.9 (11.1, 12.6)	1.07 (1.02-1.13)	1.17 (1.00-1.38)	1.37 (1.11-1.68)
Mass General Brigham					
Atrial fibrillation	1332 / 25316	2.9 (2.0, 4.1)	1.02 (0.97-1.08)	1.12 (0.95-1.33)	1.13 (0.89-1.42)
Myocardial infarction	695 / 25592	2.9 (2.0, 4.1)	0.96 (0.89-1.03)	0.87 (0.67-1.13)	0.81 (0.56-1.17)
Heart failure	1074 / 25063	2.9 (2.0, 4.1)	0.96 (0.90-1.02)	1.01 (0.83-1.23)	0.96 (0.73-1.26)
Ventricular arrhythmias	944 / 26990	3.0 (2.0, 4.2)	0.96 (0.90-1.02)	0.94 (0.75-1.16)	0.85 (0.62-1.16)
Dilated cardiomyopathy	492 / 28821	3.0 (2.1, 4.2)	1.08 (0.98-1.18)	1.09 (0.82-1.45)	1.17 (0.80-1.71)
Hypertrophic cardiomyopathy	183 / 28731	3.0 (2.1, 4.2)	1.12 (0.96-1.29)	1.13 (0.71-1.80)	0.90 (0.44-1.83)
Implantable defibrillator	152 / 28454	3.0 (2.1, 4.2)	1.06 (0.90-1.24)	1.40 (0.87-2.25)	2.20 (1.29-3.76)
*Hazard ratios obtained using Cox proportional hazards models adjusted for age, sex, and PCs 1-5					
[†] N includes all individuals without the prevalent condition at baseline					

R1.7. Consider doing a Hi-C analysis for long range chromatin interactions.

Author Response: In response to the reviewer’s comment, we have incorporated a Hi-C analysis, which we have added to **Supplementary Table 3** (reproduced below). The analysis reveals several potentially relevant chromatin interactions, including between lead variant rs56252725 on chromosome 16, and gene *MYH11*, which encodes an isoform of the myosin heavy chain which is highly expressed in LV tissue and has been associated with electrocardiogram amplitude, and between lead variant rs143973349 and gene *CCDC136*, which encodes a membrane protein and in which variants have been previously associated with dilated and hypertrophic cardiomyopathies. We discuss relevant findings in the manuscript.

Revised manuscript:

Methods (Page 9, Line 21): “We assessed for potential long-range chromatin interactions using Hi-C analysis.”

Methods (Page 10, Line 1): “We prioritized candidate genes on the basis of closest proximity to the lead variant, eQTLs, TWAS, tissue-specific expression levels, Hi-C analysis, and biologic plausibility based on previously reported data. Except where otherwise specified, all prioritized genes are supported by at least two lines of evidence.

Results (Page 16, Line 12): “Using Hi-C analysis, we observed several potentially relevant chromatin interactions, including between lead variant rs56252725 on chromosome 16 and gene *MYH11*, which encodes an isoform of the myosin heavy chain which is highly expressed in LV tissue and has been associated with electrocardiogram amplitude, and between lead variant rs143973349 (European only analysis) and gene *CCDC136*, which encodes a membrane protein and in which variants have been previously associated with dilated and hypertrophic cardiomyopathies. Detailed results of eQTL, TWAS, and Hi-C analyses are shown in **Supplementary Table 3.**”

Table 3. Bioinformatics and *in silico* functional analysis summary

Mixed-ancestry analysis												
rsID	Chr	Position (hg38)	Closest gene	GTEx v8 eQTL LV*	GTEx v8 eQTL AA*	TWAS LV	TWAS AA	Hi-C linked genes	Plausible genes within 500kb	Prioritized candidate genes	Distance to lead SNP	LV expression level
rs143800963	1	11835418	CLCN6	-	NPPA	-	-	-	CLCN6, MTHFR, NPPA, NPPB	NPPA	10291	35.81
										NPPB	22046	26.77
										CLCN6	29322	7.28
rs2255167	2	178693555	TTN	FKBP7	FKBP7	FKBP7	FKBP7	-	TTN, FKBP7, CCDC141	TTN [†]	167566	66.76
										FKBP7	229891	2.675
rs10497529	2	178975161	CCDC141	-	-	-	-	-	TTN, CCDC141	TTN [†]	449172	66.76
										CCDC141 [†]	145404	5.30
-	5	133066736	HSPA4	-	-	-	-	AFF4	HSPA4, ZCCHC10	HSPA4	14723	25.89
rs9388498	6	126552277	CENPW	-	-	-	-	-	CENPW	CENPW	212162	0.42
rs34163229	10	73647154	SYNPO2L	SYNPO2L	SYNPO2L	-	SYNPO2L	USP54	MYOZ1, PPP3CB, ANXA7, AGAP5, FUT11, SYNPO2L	SYNPO2L	2273	84.32
				-	MYOZ1	-	MYOZ1			MYOZ1	15542	1.06
				FUT11	FUT11	-	-			ANXA7	272053	50.75
				-	AGAP5	-	AGAP5			AGAP5	27133	1.74
				-	DNAJC9	-	-					
				-	DUSP8P5	-	-					
rs3729989	11	47348490	MYBPC3	PSMC3	PSMC3	-	PSMC3	-	MYBPC3, PSMC3	MYBPC3	4467	1,351
										PSMC3	91830	97.09
rs28552516	12	121592356	KDM2B	-	MORN3	-	-	ORAI1, MORN3	ORAI1, KDM2B, MORN3	ORAI1 [†]	34194	4.91
rs6598541	15	98727906	IGF1R	IGF1R	-	IGF1R	-	-	IGF1R	IGF1R	79367	5.71
rs56252725	16	14995819	PDXDC1	PDXDC1	-	PDXDC1	-	MYH11	PDXDC1, NOMO1	PDXDC1	21228	15.96
				PKD1P3	-	-	-			NOMO1	162098	15.12
				NPIPA3	-	-	-					
				NPIPA5	NPIPA5	-	-					
				RRN3	-	-	-					
				-	NOMO1	-	-			MYH11	801210	16.25
				NPIPA1	NPIPA1	-	-					
				-	AC139256.1	-	-					
				-	-	SEZ6L2	-					
rs6503451	17	45870981	MAPT	MAPT	-	-	-	-	MAPT, LRRC37A, DND1P1, MAPK8IP1P2,	KANSL1	158935	3.72
				LRRC37A4P	LRRC37A4P	-	-			MAPT	23546	4.75

				LRRC37A2	LRRC37A2	LRRC37A2	-		KANSL1, ARL17A, WNT3, CRHR1	LRRC37A	421752	0.12
				DND1P1	DND1P1	-	-					
				MAPK8IP1P2	MAPK8IP1P2	-	-					
				LINC02210	LINC02210	-	-					
				KANSL1	KANSL1	-	-					
				ARL17A	ARL17A	-	-					
				WNT3	WNT3	-	-					
				-	NSF	-	-					
rs199501	17	46784981	WNT3	WNT3	WNT3	WNT3	WNT3	-	NSF, WNT3, LRRC37A, LRRC37A2, KANSL1, ARL17A	WNT3	22475	0.41
				LRRC37A	LRRC37A	-	-			KANSL1	755065	3.72
				LRRC37A2	LRRC37A2	-	-			LRRC37A	492248	0.12
				ARL17A	ARL17A	-	-			LRRC37A2	273736	
				NSF	NSF	-	-					
				KANSL1	KANSL1	-	-					
				MAPT	-	-	-					
rs62621197	19	8605262	ADAMTS10	-	-	-	-	NFILZ	ADAMTS10, MYO1F	ADAMTS10[†]	25022	3.79
										MYO1F[‡]	84484	1.85
European-ancestry analysis												
rs143973349	7	128866182	ATP6V1F	KCP	-	KCP		CCDC136, CALU	FLNC, ATP6V1F, KCP, CCDC136, OP1SW, IRF5	FLNC[†]	35776	291.60
				-	CCDC136	-				CCDC136	435371	1.28
										KCP	4140	0.17
rs59765302	16	30018280	DOC2A	-	-	-	-	MVP	DOC2A	DOC2A	1450	0.25
*eQTLs determined using lead variants or proxy variants ($r^2 \geq 0.8$) using all-population or European-ancestry linkage disequilibrium maps from the 1,000 Genomes Project †Candidate genes prioritized on the basis of closest proximity to the lead variant, eQTLs, TWAS, tissue-specific expression levels, Hi-C analysis, and biologic plausibility based on previously reported data and are supported by at least two lines of evidence, except where otherwise specified ‡Gene prioritized at locus on the basis of strong biologic plausibility or previous association with LVM based on previous literature only §Expression levels derived from bulk tissue samples available in GTEx v8												

R1.8. Are the two loci which drop out of European analysis still borderline?

Author Response: As described above, we have excluded a small number of individuals due to withdrawal of consent, and have modified our quality control procedures in response to previous comments by this reviewer. We have therefore revised our GWAS analyses. The updated European only GWAS results are shown in **Supplementary Table 6**, which is reproduced below. In the updated analysis, when considering strong proxy variants ($r^2 > 0.8$) as representing the same locus, there are two loci which are genome-wide significant in the European-only analysis but not in the primary mixed ancestry GWAS, and a single variant that is significant in the mixed ancestry GWAS but not in the European only GWAS. As suggested, we now report p-values for the same variant in the corresponding second GWAS in both cases.

Revised manuscript:

Results (Page 15, Line 5): "In a GWAS restricted to individuals of European ancestry, 14 loci met genome-wide significance, of which 12 were either a lead variant or a strong proxy ($r^2 > 0.8$) for a lead variant in the primary GWAS (**Supplementary Table 6** and **Supplementary Figures 5-6**). The two loci unique to the European ancestry analysis were rs143973349, an insertion-deletion variant located near *FLNC*, a gene highly expressed in LV tissue and previously associated with familial hypertrophic, restrictive, and arrhythmogenic cardiomyopathies, and rs142032045, located in a gene-rich region closest to *DOC2A* and near several variants previously associated with body size.⁶⁻⁹ The

variant near *FLNC* had a suggestive association with LVMI in the primary multi-ancestry GWAS, while the variant near *DOC2A* did not ($p=3.2 \times 10^{-7}$ and $p=1.1 \times 10^{-5}$, respectively). The only variant meeting genome-wide significance in the primary mixed ancestry GWAS that was not a lead variant in the European only GWAS did have a suggestive association (rs6598541 near *IGF1R* $p=7.7 \times 10^{-8}$).

Table 6. Variants associated with CMR-derived left ventricular mass index in the European-ancestry GWAS

rsID	Chr	Position	Closest gene(s)	Function	Risk/alt allele	RAF	Beta	SE	P-value
		(hg 38)							
rs143800963	1	11835418	CLCN6	Intronic	C/A	0.95	0.98	0.17	6.8×10^{-9}
rs2255167*	2	178693555	TTN	Intronic	T/A	0.81	0.98	0.10	5.9×10^{-24}
rs10497529 [†]	5	133066736	HSPA4	Indel	CTT/C	0.72	0.47	0.09	4.4×10^{-8}
-	6	126552277	CENPW	-	G/T	0.81	-0.58	0.10	5.6×10^{-9}
rs9388498	7	128866181	ATP6V1F	Indel	TGG/T	0.82	0.57	0.10	1.0×10^{-8}
rs34163229	10	73644542	MYOZ1	Intronic	C/G	0.86	-0.61	0.11	3.7×10^{-8}
rs3729989	11	47358536	SPI1	Intronic	G/T	0.86	-0.64	0.11	1.2×10^{-8}
rs28552516	12	121592356	KDM2B	Intronic	C/T	0.85	-0.65	0.11	1.4×10^{-9}
rs6598541	16	14995819	PDXDC1	Intronic	G/A	0.75	0.53	0.10	4.5×10^{-8}
rs56252725	16	30018280	DOC2A	Indel	C/CA	0.61	-0.45	0.08	3.9×10^{-8}
rs6503451	17	45870981	MAPT	Intronic	T/C	0.67	-0.54	0.08	6.9×10^{-11}
rs199501 [‡]	17	46759287	NSF	Intronic	G/A	0.24	0.53	0.09	3.3×10^{-9}
rs62621197	19	8605262	ADAMTS10	Missense	C/T	0.96	1.24	0.21	3.7×10^{-9}

*Locus previously reported for LVM⁵

[†]Variant identified in conditional analysis conditioned on lead SNPs (beta, standard error, and p-value are adjusted)

[‡]Variant association unique to European ancestry analysis (excluding variants which are strong proxies [$r^2 > 0.8$] for primary GWAS SNPs)

[§]Association no longer observed in analysis conditioned on rs6503451

Chr = chromosome; RAF = risk allele frequency

R1.9. I could not see the number of individuals present in the analysis restricted to European ancestry? This should be very similar to the full GWAS since there were only small numbers recruited of other ethnicities in UK Biobank? What does this add?

Author Response: In response to the reviewer comment, we now report the number of individuals included in the GWAS restricted to individuals of European ancestry ($n=39,187$). Although sample size and results in the European only subset are largely similar to those of the primary mixed ancestry GWAS, which is expected, we have elected to retain both analyses since there are a few variants which differ between the two (as described in response to R1.8 from this reviewer above). Similarity between the primary mixed ancestry GWAS and the European only GWAS also provides additional evidence that the linear mixed model utilized in the primary analysis appropriately accounts for population structure.

Revised manuscript:

Methods (Page 8, Line 22): “We performed several secondary GWAS analyses. First, we performed analogous GWAS restricted to individuals of European genetic ancestry (n=39,187).”

Reviewer #2:

R2.1. This is an excellent paper taking advantage of the UK Biobank as a resource. The manuscript focuses on LV mass index as the main phenotype and identifies a number of novel SNPs in addition to detecting signals for previously identified variants. Subsequent expression analysis using GTEx data further annotates the signals. The methodology of measuring LVM through CMR is a strength.

Author Response: We appreciate the reviewer’s kind words, thoughtful review, and constructive comments.

R2.2. The authors index LVM to BSA. It is well established that this method can 'normalize' LVM in obese individuals. Another commonly used method is indexing to height^{2.7}. The authors should explore the associations signals for robustness.

Author Response: We thank the reviewer for the suggestion. In response to comments from this reviewer and others, we now perform several secondary GWAS of LVM indexed using alternative methods, including lean body mass and the 2.7th power of height. Results obtained using alternative indexing methods were broadly consistent with the primary analysis, both in terms of variants identified and effect directions. The lead variants for GWAS of LVM indexed using the 2.7th power of height are shown in **Supplementary Table 7**. A summary of association results for the lead variants identified in the primary GWAS tested across the varying indexing methods is provided in **Supplementary Table 12**. Both tables are reproduced below.

Revised manuscript:

Methods (Page 8, Line 22): “We performed several secondary GWAS analyses. First, we performed analogous GWAS restricted to individuals of European genetic ancestry (n=39,187).”

Results (Page 15, Line 18): “Results of secondary GWAS analyses, including rank-based inverse normal transformed LVMI, LVMI indexed using the 2.7th power of height, LVMI indexed using lean body mass, LVMI with exclusions for prevalent myocardial infarction and heart failure, and unindexed LVM adjusted for height and weight, are shown in **Supplementary Tables 7-11**. Results obtained using alternative indexing methods were broadly consistent with the primary analysis, both in terms of variants identified and effect directions. A summary of association results for the lead variants identified in the primary GWAS tested across varying indexing methods is shown in **Supplementary Table 12**.”

Table 7. Variants associated with CMR-derived left ventricular mass indexed using the 2.7th power of height

rsID	Chr	Position	Closest gene(s)	Function	Risk/alt allele	RAF	Beta	SE	P-value
		(hg 38)							
rs2255167*	2	178693555	TTN	Intronic	T/A	0.81	0.41	0.047	1.4x10 ⁻¹⁸
rs571173399	5	133106107	HSPA4	Intronic	T/G	0.77	0.25	0.044	1.1x10 ⁻⁸
rs28552516	12	121592356	KDM2B	Intronic	C/T	0.85	-0.30	0.052	4.0x10 ⁻⁹
rs1421085	16	53767042	FTO	Intronic	T/C	0.60	-0.25	0.038	4.0x10 ⁻¹¹
rs6503451	17	45870981	MAPT	Intronic	T/C	0.67	-0.26	0.041	1.3x10 ⁻¹⁰
rs62071449†	17	46613342	NSF	Intronic	G/A	0.81	-0.29	0.050	7.3x10 ⁻⁹

*Locus previously reported for LVM⁵
†Association no longer observed in analysis conditioned on rs6503451
Chr., chromosome; RAF, risk allele frequency

Table 12. Associations with CMR-derived left ventricular mass across varying indexing methods

rsID*	Chr	Position (hg38)	Closest gene(s)	Body surface area		Body surface area (rank-inverse normal transformation)		2.7 th power of height		Lean body mass	
				Beta†	P-value	Beta†	P-value	Beta†	P-value	Beta†	P-value
rs143800963	1	11835418	CLCN6	0.078	4.2x10 ⁻⁹	0.077	6.5x10 ⁻⁹	0.073	2.5x10 ⁻⁷	0.082	3.9x10 ⁻⁸
rs2255167	2	178693555	TTN	0.079	3.2x10 ⁻²⁶	0.079	1.5x10 ⁻²⁶	0.071	1.4x10 ⁻¹⁸	0.094	7.2x10 ⁻²⁹
rs10497529‡	2	178975161	CCDC141	0.100	3.0x10 ⁻¹⁰	0.100	6.9x10 ⁻¹⁰	0.095	6.0x10 ⁻⁸	0.120	8.8x10 ⁻¹¹
-	5	133066736	HSPA4	0.041	1.6x10 ⁻⁹	0.041	1.1x10 ⁻⁹	0.040	2.7x10 ⁻⁸	0.049	4.0x10 ⁻¹¹
rs9388498	6	126552277	CENPW	-0.045	4.1x10 ⁻⁹	-0.045	3.4x10 ⁻⁹	-0.042	3.4x10 ⁻⁷	-0.045	2.1x10 ⁻⁷
rs34163229	10	73647154	SYNPO2L	-0.049	1.0x10 ⁻⁸	-0.049	6.6x10 ⁻⁹	-0.049	1.1x10 ⁻⁷	-0.051	9.8x10 ⁻⁸
rs3729989	11	47348490	MYBPC3	-0.050	1.8x10 ⁻⁸	-0.051	8.1x10 ⁻⁹	-0.047	4.9x10 ⁻⁷	-0.049	8.0x10 ⁻⁷
rs28552516	12	121592356	KDM2B	-0.047	1.5x10 ⁻⁸	-0.050	2.5x10 ⁻⁹	-0.052	4.0x10 ⁻⁹	-0.054	7.3x10 ⁻⁹
rs6598541	15	98727906	IGF1R	-0.034	4.6x10 ⁻⁸	-0.035	3.9x10 ⁻⁸	-0.035	1.8x10 ⁻⁷	-0.043	1.0x10 ⁻⁹
rs56252725	16	14995819	PDXDC1	0.044	3.7x10 ⁻⁹	0.044	3.0x10 ⁻⁹	0.039	9.0x10 ⁻⁷	0.045	8.5x10 ⁻⁸
rs6503451	17	45870981	MAPT	-0.042	1.1x10 ⁻¹⁰	-0.043	4.8x10 ⁻¹¹	-0.045	1.3x10 ⁻¹⁰	-0.041	1.9x10 ⁻⁸
rs199501	17	46785247	WNT3	0.043	1.1x10 ⁻⁹	0.042	3.1x10 ⁻⁹	0.043	1.0x10 ⁻⁸	0.039	6.1x10 ⁻⁷
rs62621197	19	8605262	ADAMTS10	0.091	2.9x10 ⁻⁸	0.088	6.6x10 ⁻⁸	0.060	4.9x10 ⁻⁴	0.110	7.6x10 ⁻¹⁰

*Variants shown are those significant in the primary GWAS
†All betas are presented per 1-standard deviation increase to facilitate comparisons
‡Variant identified in conditional analysis conditioned on lead SNPs using respective indexing method (beta, standard error, and p-value are adjusted)

R2.3. The authors perform a mendelian randomization for hypertension. Diabetes is another well-established risk factor and might warrant further examination.

Author Response: In response to the reviewer's suggestion we have added an additional Mendelian randomization analysis for diabetes.

Revised manuscript:

Methods (Page 12, Line 5): As a form of validation of our LVM estimation, we sought to identify evidence of known causal associations between elevated blood pressure and increased LVM.³ We therefore conducted two-sample Mendelian randomization (MR) within individuals of genetic European ancestry in the UK Biobank sample. We performed analogous analyses for diabetes. Genetic instruments for systolic blood pressure (SBP) and diastolic blood pressure (DBP) were derived from a recent GWAS.¹⁰ Utilizing an 865 SNP instrument for SBP and DBP, we prioritized inverse-variance weighted (IVW) meta-analyses of the effect of each SNP on CMR-derived LVMI (and LVM) divided by the effect of the same SNP on SBP and DBP, respectively. We performed an analogous procedure using a 337 SNP instrument for diabetes.¹¹ Weighted median and MR-Egger analyses were performed secondarily to address potential invalid instruments and directional pleiotropy. Further details of the MR analysis are provided in the **Supplementary Methods**.

Results (Page 18, Line 19): To assess for potential causal associations between blood pressure and CMR-derived LVMI, we performed MR analyses using genetic instruments for SBP and DBP among individuals of European ancestry. We performed analogous analyses for diabetes. In an inverse variance weighted two-sample MR, a 1-SD increase in genetically mediated SBP was associated with a 0.27g/m² increase in CMR-derived LVMI (95% CI, 0.23-0.31, p=1.75x10⁻⁴¹), and a 1-SD increase in genetically mediated DBP was associated with a 0.32g/m² increase in CMR-derived LVMI (95% CI, 0.25-0.39, p=1.64x10⁻²⁰). A 1-SD increase in genetically mediated risk of diabetes was associated with a 0.31g/m² increase in CMR-derived LVMI (95% CI, 0.05-0.56, p=0.018). Weighted median and MR-Egger analyses demonstrated similar results for SBP and DBP, but associations with diabetes were no longer significant (weighted median: 0.19g/m², 95% CI -0.15-0.53, p=0.26; MR-Egger: 0.15g/m², 95% CI -0.36-0.66, p=0.56). MR-Egger analyses suggested no substantive directional pleiotropy in the SBP, DBP, and diabetes instruments (intercept 0.01, p=0.38 for SBP; intercept -0.02, p=0.04 for DBP; intercept=0.01, p=0.50 for diabetes). MR results were similar using unindexed LVM (**Supplementary Table 16**). MR plots are shown in **Supplementary Figure 7**.

Table 16. Mendelian randomization for LVMI and LVM

Inverse-variance weighted						
Phenotype	LVMI Beta (95% CI)*	LVMI p		LVM Beta (95% CI)*	LVM p	
Systolic blood pressure	0.27 (0.23-0.31)	1.75x10 ⁻⁴¹		0.44 (0.35-0.53)	8.16x10 ⁻²³	
Diastolic blood pressure	0.32 (0.25-0.39)	1.64x10 ⁻²⁰		0.54 (0.39-0.69)	1.74x10 ⁻¹²	
Diabetes	0.31 (0.05-0.56)	0.018		0.62 (0.004-1.23)	0.048	
Weighted median						
Phenotype	LVMI Beta (95% CI)*	LVMI p		LVM Beta (95% CI)*	LVM p	
Systolic blood pressure	0.28 (0.23-0.33)	6.65x10 ⁻²⁸		0.49 (0.37-0.60)	1.93x10 ⁻¹⁷	
Diastolic blood pressure	0.33 (0.25-0.42)	3.03x10 ⁻¹⁴		0.59 (0.40-0.78)	7.82x10 ⁻¹⁰	
Diabetes	0.19 (-0.15-0.53)	0.26		0.02 (-0.73-0.77)	0.96	
MR-Egger						
Phenotype	LVMI Beta (95% CI)*	LVMI p	LVMI Intercept (p-value)†	LVM Beta (95% CI)*	LVM p	LVM Intercept (p-value)†
Systolic blood pressure	0.24 (0.16-0.31)	3.23x10 ⁻¹⁰	0.01 (0.38)	0.38 (0.21-0.55)	1.07x10 ⁻⁵	0.02 (0.40)
Diastolic blood pressure	0.42 (0.30-0.54)	1.56x10 ⁻¹²	-0.02 (0.04)	0.76 (0.51-1.02)	6.87x10 ⁻⁹	-0.03 (0.04)
Diabetes	0.15 (-0.36-0.66)	0.56	0.01 (0.50)	-0.33 (-1.56-0.89)	0.59	0.06 (0.08)
*Beta estimates represent the expected causal effect per 1-standard deviation increase in the respective risk factor (phenotype) on LVMI and LVM, respectively						
†A non-zero intercept suggests the presence of directional pleiotropy, where a significant p-value indicates a statistically significant difference from zero						

Figure 7. Two-sample Mendelian randomization plots

Depicted are scatterplots depicting results of two-sample Mendelian randomization. Each point is a genetic variant, the x-axis depicts strength of association between the variant and the exposure (i.e., systolic blood pressure, diastolic blood pressure, and diabetes, as labeled above each plot). The y-axis depicts strength of association between the variant and the outcome (i.e., left ventricular mass index in the top panels, and left ventricular mass in the bottom panels). Each plot depicts the result of inverse variance weighted regression (IVW, blue) and MR-Egger regression (red). A red line crossing the origin (y-intercept close to zero) suggests absence of substantial directional pleiotropy in the genetic instrument.

R2.4. The outcomes analysis is interesting and potentially clinically important. However, this will require further descriptions. First, the selected outcomes are diverse. While AF and HF appear as expected progression of LVH, DCM or HCM could be understood from a genetic perspective as a more monogenic disease. Are the authors hereby suggesting that HCM can be also caused by a combination of variants from a PRS, in addition to the well-established mutations in HCM genes? This needs further clarification. Same holds for DCM. Furthermore, are patients with known mutations in genes for DCM or HCM excluded and if not how could this affect the analysis? It could appear that even a small number of patients with mutations in known HCM genes could impact the results.

Author Response: We agree it would be useful to clarify our rationale in the PRS outcomes analysis. We elected not to exclude individuals with pathogenic or likely pathogenic variants for HCM or DCM for the following reasons: a) a substantial proportion of individuals with clinically confirmed HCM and DCM have no causal variant identified,^{12,13} b) recent evidence suggests that polygenic background may play an

important role in disease development even among individuals carrying pathogenic mutations,¹⁴ and c) rare variant information is not available in all individuals in our UK Biobank sample, and was not available for individuals in the MGB replication sample. In response to the reviewer’s comments, we have clarified our methods and rationale in the Methods section. In addition, in **Table 4** we now report the number of events occurring in individuals carrying high confidence loss-of-function, deleterious missense, and known pathogenic or likely pathogenic variants in HCM and DCM genes as catalogued in ClinVar as of 2/9/2021. A list of variants is provided in **Supplementary Table 2**.

Revised manuscript:

Methods (Page 13, Line 3): “For the PRS outcomes analyses, we did not exclude individuals with pathogenic or likely pathogenic variants for HCM or DCM for the following reasons: a) a substantial proportion of individuals with clinically confirmed HCM and DCM have no causal variant identified,^{12,13} b) recent evidence suggests that polygenic background may play an important role in disease development even among individuals carrying mutations,¹⁴ and c) rare variant information is not available in all individuals in our UKBB or MGB replication samples, although we did tabulate carrier status of high confidence loss-of-function, deleterious missense, and known pathogenic or likely pathogenic variants in HCM and DCM genes as catalogued in ClinVar as of 2/9/2021 (see **Supplementary Methods** and **Supplementary Table 2**).”

Supplementary Methods:

Tabulation of pathologic rare variants for HCM and DCM

To assess the frequency of pathologic rare variants among individuals with incident HCM and DCM events, we tabulated carrier status of known pathogenic or likely pathogenic variants in HCM and DCM genes as catalogued in ClinVar as of 2/9/2021. In addition, we also identified high confidence loss-of-function variants using LOFTEE,¹⁵ a plug-in of VEP.¹⁶ We also included deleterious missense variants¹⁷ using 30 in-silico prediction tools presented in v4.1a of the dbnsfp database.¹⁸ A list of variants is shown in **Supplementary Table 2**.

Table 4. Associations between LVMI PRS and incident disease

	N events / N total [†]	Follow-up, yrs (Q1, Q3)	Hazard ratio for covariate (95% CI)*		
			PRS (per 1 SD)	PRS (90 th percentile)	PRS (95 th percentile)
UK Biobank					
Atrial fibrillation	25050 / 435917	11.8 (11.0, 12.6)	1.02 (1.00-1.03)	1.05 (1.00-1.09)	1.06 (1.00-1.12)
Myocardial infarction	13405 / 432044	11.8 (11.0, 12.6)	1.02 (1.00-1.04)	1.08 (1.02-1.14)	1.09 (1.02-1.18)
Heart failure	13540 / 440590	11.9 (11.0, 12.6)	1.04 (1.02-1.05)	1.07 (1.02-1.13)	1.07 (0.99-1.15)
Ventricular arrhythmias	4882 / 442295	11.9 (11.1, 12.6)	1.06 (1.03-1.09)	1.15 (1.05-1.26)	1.21 (1.07-1.36)
Dilated cardiomyopathy [‡]	1023 / 443013	11.9 (11.1, 12.6)	1.09 (1.03-1.16)	1.24 (1.03-1.50)	1.25 (0.96-1.61)
Hypertrophic cardiomyopathy [‡]	420 / 443150	11.9 (11.1, 12.6)	1.13 (1.02-1.24)	1.07 (0.78-1.48)	1.18 (0.77-1.80)
Implantable defibrillator	1444 / 443216	11.9 (11.1, 12.6)	1.06 (1.01-1.12)	1.12 (0.95-1.32)	1.26 (1.02-1.56)
Mass General Brigham					
Atrial fibrillation	1332 / 25316	2.9 (2.0, 4.1)	1.01 (0.95-1.06)	1.02 (0.85-1.22)	1.03 (0.80-1.31)
Myocardial infarction	695 / 25592	2.9 (2.0, 4.1)	0.99 (0.92-1.06)	0.97 (0.74-1.25)	0.71 (0.47-1.07)
Heart failure	1074 / 25063	2.9 (2.0, 4.1)	0.97 (0.91-1.03)	1.18 (0.97-1.42)	1.00 (0.76-1.33)

Ventricular arrhythmias	944 / 26990	3.0 (2.0, 4.2)	0.99 (0.93-1.05)	1.00 (0.81-1.24)	1.03 (0.76-1.38)
Dilated cardiomyopathy	492 / 28821	3.0 (2.1, 4.2)	1.06 (0.97-1.16)	1.27 (0.97-1.67)	1.06 (0.70-1.59)
Hypertrophic cardiomyopathy	183 / 28731	3.0 (2.1, 4.2)	1.14 (0.98-1.32)	1.04 (0.64-1.69)	0.82 (0.38-1.75)
Implantable defibrillator	152 / 28454	3.0 (2.1, 4.2)	1.05 (0.89-1.24)	1.75 (1.12-2.74)	1.69 (0.91-3.12)

*Hazard ratios obtained using Cox proportional hazards models adjusted for age, sex, and principal components 1-5

†N includes all individuals without the prevalent condition at baseline

‡ Includes n=20 events with high confidence loss-of-function, deleterious missense, known pathogenic or likely pathogenic variant for HCM, and n=50 events with high confidence loss-of-function, deleterious missense, known pathogenic or likely pathogenic rare variant for DCM (see text and **Supplementary Table 2**)

CI = confidence interval, PRS = polygenic risk score, Q1 = quartile 1, Q3 = quartile 3, SD = standard deviation

R2.5. Furthermore, the authors describe a progression to HF. If a PRS could be suggested as a prognostic marker, this will also need further detail. Obviously, a previous reduction on LV function is likely to further accelerate the progression to HF, as its treatment. The authors might want to consider a more detailed analysis related to HF which includes LV function parameters at baseline as well as medication use to explore the role of the PRS. Without this information, the results are potentially interesting, however a potential clinical use might be less clear.

Author Response: We submit that our PRS outcomes analysis provides evidence that the genetic variation underlying increased LVM may be clinically relevant, and that our findings warrant future research to clarify the potential utility of a polygenic indicator of LVM to improve identification of individuals at greater risk of incident cardiomyopathy. We agree with the reviewer that such an analysis would require detailed consideration of other baseline LV structural parameters as well as additional clinical factors (e.g., medication use). We submit that such an analysis is outside the scope of the current work. In response to the reviewer's comment, we have modified the Discussion to reflect this point.

Revised manuscript:

Discussion (Page 22, Line 15): "Overall, our findings provide evidence that the genetic variation underlying increased LVM may be clinically relevant, and highlight the need for future research to evaluate the potential utility of a polygenic predictor of LVM to improve identification of individuals at risk of incident cardiomyopathy."

Reviewer #3:

R3.1. In this study the authors conduct a genome-wide association study (GWAS) of left-ventricular mass (LVM) as predicted using a deep learning algorithm. The 12 loci highlighted by this approach appear to be biologically plausible and the application of a polygenic risk score using these findings is associated with risk of cardiomyopathies in independent samples.

For the most part, this study is appropriately conducted and the manuscript is well written. I have included several comments below to try and help refine this work. In particular, I found the eQTL/TWAS and MR components to be the weakest parts of the manuscript.

Author Response: We thank the reviewer for their careful read and constructive feedback. We respond to specific comments below.

R3.2. The main question I had regarding the GWAS results is how many novel associations have been identified by previous GWAS of heart structure phenotypes? For example, several authors on this work previously conducted a GWAS of MRI-derived left ventricular phenotypes

in the UK Biobank subset of participants:

<https://www.nature.com/articles/s41467-01439720-15823-7>

This previous work is referenced in the current manuscript (ref 21) - although I found it strange that this is in relation to the source of eQTL data used rather than in relation to the loci highlighted by this new study.

I would ask that the authors provide a summary of how the 12 loci associated with LVM in this new study relate to the various heart structure phenotypes using results from the study linked above as all these summary statistics are available. A summary table would be helpful for this to compare side by side estimates. A quick skim looks like there will be overlapping GWAS loci (e.g. *FLNC*) - so this will help give the reader an idea about how many have been previously reported to play a role in heart structure phenotypic variation.

Author Response: We thank the reviewer for this suggestion, and in response we now include a summary of our lead variants in the context of prior associations with LV structural and functional traits (both by Pirruccello et al.¹⁹, and Aung et al.⁵). A few of the loci we identified have been previously associated with other LV traits. As suggested, we now present these findings in new **Supplementary Table 13**, and have added relevant discussion.

Revised manuscript:

Methods (Page 10, Line 7): “To assess whether the variants we identified in association with LVMI have been previously associated with other LV traits, we compared our loci to those reported to have genome-wide associations with other LV traits reported in prior analyses by Pirruccello et al.¹⁹ and Aung et al.⁵ For this analyses, we tabulated all associations including the same variant, a variant serving as a strong proxy ($r^2 \geq 0.80$), or a variant mapping to the same candidate gene.”

Results (Page 17, Line 10): “We assessed whether the significant loci we identified have been previously implicated in association with LV traits. In addition to a prior association with LVM, rs2255167 near *TTN* has been previously associated with LV end diastolic volume, LV end systolic volume, and LV ejection fraction. Variant rs6503451 near *MAPT* and rs199501 near *WNT3* are located at regions previously associated with LV end-systolic volume. In the European-only analysis, variant rs143973349 near *FLNC* has also been associated with LV end-systolic volume, as well as LV ejection fraction. Details of lead variants and their associations in prior analyses are shown in **Supplementary Table 13.**”

Discussion (Page 20, Line 2): “CMR-derived LVMI was strongly associated with variation at SNP rs2255167, located on the large sarcomeric protein *TTN* and previously associated with LV mass,⁵ as well as LV volumes and ejection fraction.¹⁹”

Discussion (Page 21, Line 11): “One of the loci detected in the European ancestry sub-analysis (and suggestive in the primary analysis), *FLNC*, encodes filamin C, an actin-related protein associated with familial HCM,⁷ restrictive cardiomyopathy,⁸ arrhythmogenic cardiomyopathy,⁶ and LV contractile function.¹⁹”

Table 13. Prior associations of lead variants with left ventricular traits

rsID	Chr	Position (hg38)	Prioritized candidate gene(s)	Beta (SE)	P-value	Previous associations*
Primary association analysis						
rs143800963	1	11835418	CLCN6 NPPA NPPB	0.95 (0.16)	4.2x10 ⁻⁹	
rs2255167	2	178693555	TTN FKBP7	0.97 (0.09)	3.2x10 ⁻²⁶	LVM (rs2255167 [†]) LVEDV (rs2042995 [†]) LVESV (rs2042995 [†] , rs2562845 [‡]) LVESVi (rs2562845 [‡]) LVEF (rs2042995 [†] , rs2562845 [‡])
rs10497529	2	178975161	TTN CCDC141	1.28 (0.20)	2.2x10 ⁻⁹	
-	5	133066736	HSPA4	0.50 (0.08)	1.6x10 ⁻⁹	
rs9388498	6	126552277	CENPW	-0.55 (0.10)	4.1x10 ⁻⁹	
rs34163229	10	73647154	SYNPO2L MYOZ1 ANXA7 PPP3CB AGAP5	-0.60 (0.10)	1.0x10 ⁻⁸	
rs3729989	11	47348490	MYBPC3 PSMC3	-0.61 (0.11)	1.8x10 ⁻⁸	
rs28552516	12	121592356	ORAI1	-0.58 (0.10)	1.5x10 ⁻⁸	
rs6598541	15	98727906	IGF1R	-0.42 (0.08)	4.6x10 ⁻⁸	
rs56252725	16	14995819	PDXDC1 NOMO1 MYH11	0.54 (0.09)	3.7x10 ⁻⁹	
rs6503451	17	45870981	KANSL1 MAPT LRRC37A LRRC37A2	-0.52 (0.08)	1.1x10 ⁻¹⁰	LVESV (rs242562 [†]) LVESVi (rs242562 [†])
rs199501	17	46785247	WNT3 KANSL1 LRRC37A LRRC37A2	0.55 (0.09)	1.1x10 ⁻⁹	LVESV (rs242562 [†]) LVESVi (rs242562 [†])
rs62621197	19	8605262	ADAMTS10	1.11 (0.20)	2.9x10 ⁻⁸	
European ancestry analysis						
rs143973349	7	128866181	FLNC CCDC136 KCP	0.57 (0.10)	1.0x10 ⁻⁰⁸	LVESV (rs34373805 [†]) LVESVi (rs34373805 [†]) LVEF (rs3807309 [†])
rs142032045	16	30018280	DOC2A	-0.45 (0.08)	3.9x10 ⁻⁰⁸	
*Traits listed for prior associations with the respective lead variant, a strong proxy ($r^2 \geq 0.80$), or a variant mapped to the same gene. The variant implicated in the prior analysis is listed in parenthesis. [†]Association reported in Aung et al.⁵ [‡]Association reported in Pirruccello et al.¹⁹ Chr = chromosome; LVEF = left ventricular ejection fraction; LVESV = left ventricular end-systolic volume; LVESVi = left ventricular end-systolic volume index; SE = standard error						

R3.3. Did the authors take any further steps to account for the non-European subset of participants in their GWAS? How did they account for the allele frequency differences amongst

participants? Most of the literature I am familiar with has conducted GWAS of participants of different ancestries separately before meta-analysing findings together. I think it would be good for the authors to justify this aspect of their study design with references to previous studies who have undertaken a similar approach. I appreciate that the European only GWAS results are available to the reader, but I think the decision to effectively ignore ancestry in the primary GWAS would be worthwhile justifying as some readers will likely be unfamiliar with this approach.

Author Response: We conducted a mixed-ancestry GWAS using BOLT-LMM,²⁰ which accounts for ancestral heterogeneity, cryptic population structure, and sample relatedness by fitting a linear mixed model with a Bayesian mixture prior as a random effect.²⁰ Previous evidence supports the use of LMM approaches to perform GWAS of admixed populations, which may provide favorable statistical power.^{21–23} In response to the reviewer’s comment, we now provide more detailed description of our methods and include references to studies utilizing the same or similar approaches.^{19,23,24} Furthermore, we submit that our observation of a genomic control factor of 1.15 with an LD score regression intercept of 1.00 in the primary analysis is reassuring against inflation on account of residual population stratification, and that the presence of substantively similar findings in the European-only analysis supports the validity of the primary GWAS.

Revised manuscript:

Methods (Page 8, Line 1): “To identify common genetic variation associated with CMR-derived LVM, we performed a GWAS of indexed LVM using BOLT-LMM,²⁰ which accounts for ancestral heterogeneity, cryptic population structure, and sample relatedness by fitting a linear mixed model with a Bayesian mixture prior as a random effect.^{19,23,24}”

Methods. Supplemental methods

Genome-wide association study of LVMI

We performed a genome-wide association study (GWAS) using CMR-derived LV mass index (LVMI), estimated using our deep learning segmentation model, as the variable of interest. The GWAS was performed using BOLT-LMM,²⁰ which accounts for ancestral heterogeneity, cryptic population structure, and sample relatedness by fitting a linear mixed model with a Bayesian mixture prior as a random effect.²⁰ Previous evidence supports the use of LMM approaches to perform GWAS of admixed populations, which may provide favorable statistical power.^{21–23} Similar approaches have been taken previously.^{19,23,24} In the model, CMR-derived LVMI was the outcome of interest, with age at MRI, sex, array platform, and the first five principal components of genetic ancestry as covariates.

R3.4. The eQTL/TWAS section seems like quite a light touch in terms of how much of the manuscript discusses this analysis. How well do the authors believe this strategy helped map GWAS signals to causal genes? For example, in supplementary table 2 the TTN locus has 'FKBP7' in the corresponding 4 columns relating to eQTL/TWAS and yet the authors prioritise TTN based on 'strong biologic plausibility'. Does this not diminish the utility of this SNP to gene mapping strategy i.e. just ignoring the findings? Can the authors comment on loci where they believe this approach was worthwhile?

Author Response: In response to comments by this reviewer and an additional reviewer, we have revised our post-GWAS analyses, including the addition of Hi-C analyses. Based on this reviewer's feedback, we have expanded our reporting of the downstream analyses and highlight areas where eQTL/TWAS/Hi-C supported candidate gene selection based on proximity or biological plausibility, as well as cases where eQTL/TWAS/Hi-C assisted in the identification of plausible candidate genes that may otherwise not have been prioritized. We acknowledge that there were a limited set of instances where the candidate gene was prioritized solely based on prior associations with LVM and/or strong biological plausibility, including *TTN* which has been previously associated with LVM in GWAS,⁵ and both *MYBPC3*^{25,26} and *FLNC*,⁷ which are known HCM genes which are highly expressed in LV tissue.

Revised manuscript:

Results (Page 16, Line 6): "In total, of the 12 independent lead SNPs, eight (or their proxies at $r^2 \geq 0.8$) were significant eQTLs in LV and/or AA tissue samples (**Figure 3**). The locus including variant rs143973349 unique to the European ancestry analysis also included eQTLs for LV and AA tissue. For a significant proportion of candidate genes, expression was identified in both LV and AA tissue samples. We then performed TWAS and identified 6 genes across 5 loci where predicted expression was associated with LVM. Each of the genes implicated by TWAS was also an eQTL for either LV or AA (**Figure 3**). Using Hi-C analysis, we observed several potentially relevant chromatin interactions, including between lead variant rs56252725 on chromosome 16 and gene *MYH11*, which encodes an isoform of the myosin heavy chain which is highly expressed in LV tissue and has been associated with electrocardiogram amplitude, and between lead variant rs143973349 (European only analysis) and gene *CCDC136*, which encodes a membrane protein and in which variants have been previously associated with dilated and hypertrophic cardiomyopathies. Detailed results of eQTL, TWAS, and Hi-C analyses are shown in **Supplementary Table 3**.

Probable candidate genes at each locus of interest are summarized in **Figure 3**. In several cases, the closest gene was additionally supported by either eQTL or TWAS prioritization, including *SYNPO2L* near rs56252725, *IGF1R* near rs6598541, *PDXDC1* near rs56252725, *MAPT* near rs6598541, and *WNT3* near rs199501. In selected instances, downstream analyses prioritized alternative genes, including *NPPA* near rs143800963 and *ORA11* near rs28552516, with both genes having substantial expression in LV tissue. Selected genes prioritized solely based on strong biologic plausibility or previous associations with LVM included *TTN* near rs255167, *MYBPC3* near rs3729989, and *FLNC* near rs143973349 (EUR only subset). *TTN*, *MYBPC3*, and *FLNC* are also substantially expressed in LV tissue (**Supplementary Table 3**)."

R3.5. I also think that the Mendelian randomization analysis is the weakest section of the paper. Findings from this analysis are not mentioned in the discussion at all. I also have no idea what the term 'genetically mediated' means. Are the authors able to clarify?

Overall, I don't think this section really adds much to the paper. I don't believe this is a compelling 'validation' of the LVM measure as the authors suggest and think that the PRS analysis is a lot stronger in this respect...I'm struggling to see the value added by the MR.

If the authors feel very strongly about retaining it then to start with I would recommend changing the interpretation of findings to 'genetically predicted' rather than 'mediated'. The instrument selection is also quite strange...the authors use blood pressure findings from a study which is over a decade old. Maybe they are concerned about sample overlap in the UKB cohort but I

don't think this will be too detrimental in terms of overfitting given that LVM is based on a subset of UKB. Is there also not a more recent blood pressure GWAS since 2011 which didn't include UKB?

Author Response: In response to comments by this reviewer and an additional reviewer, we have substantially revised our Mendelian randomization analyses. In response to a specific comment by another reviewer, we have added a Mendelian randomization analysis for diabetes. As suggested, we have also updated our blood pressure instrument. We have modified our interpretation to the suggested language of 'genetically predicted'. We now present all findings in **Supplementary Table 16**, show corresponding MR plots in **Supplemental Figure 7**, and discuss our findings in greater context in the Discussion.

Revised manuscript:

Methods (Page 12, Line 5): As a form of validation of our LVM estimation, we sought to identify evidence of known causal associations between elevated blood pressure and increased LVM.³ We therefore conducted two-sample Mendelian randomization (MR) within individuals of genetic European ancestry in the UK Biobank sample. We performed analogous analyses for diabetes. Genetic instruments for systolic blood pressure (SBP) and diastolic blood pressure (DBP) were derived from a recent GWAS.¹⁰ Utilizing an 865 SNP instrument for SBP and DBP, we prioritized inverse-variance weighted (IVW) meta-analyses of the effect of each SNP on CMR-derived LVMI (and LVM) divided by the effect of the same SNP on SBP and DBP, respectively. We performed an analogous procedure using a 337 SNP instrument for diabetes.¹¹ Weighted median and MR-Egger analyses were performed secondarily to address potential invalid instruments and directional pleiotropy. Further details of the MR analysis are provided in the **Supplementary Methods**.

Results (Page 18, Line 19): To assess for potential causal associations between blood pressure and CMR-derived LVMI, we performed MR analyses using genetic instruments for SBP and DBP among individuals of European ancestry. We performed analogous analyses for diabetes. In an inverse variance weighted two-sample MR, a 1-SD increase in genetically mediated SBP was associated with a 0.27g/m² increase in CMR-derived LVMI (95% CI, 0.23-0.31, p=1.75x10⁻⁴¹), and a 1-SD increase in genetically mediated DBP was associated with a 0.32g/m² increase in CMR-derived LVMI (95% CI, 0.25-0.39, p=1.64x10⁻²⁰). A 1-SD increase in genetically mediated risk of diabetes was associated with a 0.31g/m² increase in CMR-derived LVMI (95% CI, 0.05-0.56, p=0.018). Weighted median and MR-Egger analyses demonstrated similar results for SBP and DBP, but associations with diabetes were no longer significant (weighted median: 0.19g/m², 95% CI -0.15-0.53, p=0.26; MR-Egger: 0.15g/m², 95% CI -0.36-0.66, p=0.56). MR-Egger analyses suggested no substantive directional pleiotropy in the SBP, DBP, and diabetes instruments (intercept 0.01, p=0.38 for SBP; intercept -0.02, p=0.04 for DBP; intercept=0.01, p=0.50 for diabetes). MR results were similar using unindexed LVM (**Supplementary Table 16**). MR plots are shown in **Supplemental Figure 7**.

Discussion (Page 22, Line 13): Consistent with expectations,^{3,27} using Mendelian randomization analyses, we observed associations between genetically predicted blood pressure and diabetes risk with greater LVM. Overall, our findings provide evidence that the genetic variation underlying increased LVM may be clinically relevant, and highlight the need for future research to evaluate the potential utility of a polygenic predictor of LVM to improve identification of individuals at risk of incident cardiomyopathy.

Table 16. Mendelian randomization for LVMI and LVM

Inverse-variance weighted						
Phenotype	LVMI Beta (95% CI)*	LVMI p	LVM Beta (95% CI)*	LVM p		
Systolic blood pressure	0.29 (0.25-0.33)	6.12x10 ⁻⁴³	0.48 (0.39-0.57)	4.38x10 ⁻²⁴		
Diastolic blood pressure	0.34 (0.27-0.41)	5.25x10 ⁻²¹	0.59 (0.43-0.75)	2.57x10 ⁻¹³		
Diabetes	0.34 (0.09-0.60)	8.9x10 ⁻³	0.66 (0.04-1.27)	0.036		
Weighted median						
Phenotype	LVMI Beta (95% CI)*	LVMI p	LVM Beta (95% CI)*	LVM p		
Systolic blood pressure	0.28 (0.22-0.33)	1.20x10 ⁻²³	0.44 (0.32-0.57)	6.06x10 ⁻¹³		
Diastolic blood pressure	0.39 (0.30-0.48)	2.17x10 ⁻¹⁶	0.55 (0.35-0.75)	1.09x10 ⁻⁷		
Diabetes	0.33 (-0.02-0.68)	0.067	-0.060 (-0.84-0.72)	0.88		
MR-Egger						
Phenotype	LVMI Beta (95% CI)*	LVMI p	LVMI Intercept (p-value)†	LVM Beta (95% CI)*	LVM p	LVM Intercept (p-value)†
Systolic blood pressure	0.26 (0.18-0.33)	1.47x10 ⁻¹⁰	0.01 (0.38)	0.41 (0.23-0.59)	6.60x10 ⁻⁶	0.02 (0.36)
Diastolic blood pressure	0.45 (0.33-0.57)	4.96x10 ⁻¹³	-0.02 (0.03)	0.85 (0.58-1.12)	1.04x10 ⁻⁹	-0.04 (0.02)
Diabetes	0.22 (-0.28-0.73)	0.39	0.01 (0.60)	-0.18 (1.40-1.03)	0.77	0.05 (0.12)

*Beta estimates represent the expected causal effect per 1-standard deviation increase in the respective risk factor (phenotype) on LVMI and LVM, respectively
†A non-zero intercept suggests the presence of directional pleiotropy, where a significant p-value indicates a statistically significant difference from zero

Figure 7. Two-sample Mendelian randomization plots

Depicted are scatterplots depicting results of two-sample Mendelian randomization. Each point is a genetic variant, the x-axis depicts strength of association between the variant and the exposure (i.e., systolic blood pressure, diastolic blood pressure, and diabetes, as labeled above each plot). The y-axis depicts strength of association between the variant and the outcome (i.e., left ventricular mass index in the top panels, and left ventricular mass in the bottom panels).

Each plot depicts the result of inverse variance weighted regression (IVW, blue) and MR-Egger regression (red). A red line crossing the origin (y-intercept close to zero) suggests absence of substantial directional pleiotropy in the genetic instrument.

References

1. Khurshid, S. *et al.* Deep learning to estimate cardiac magnetic resonance–derived left ventricular mass. *Cardiovascular Digital Health Journal* S2666693621000232 (2021) doi:10.1016/j.cvdhj.2021.03.001.
2. Khurshid, S. *et al.* Deep learning to estimate cardiac magnetic resonance–derived left ventricular mass. *Cardiovascular Digital Health Journal* S2666693621000232 (2021) doi:10.1016/j.cvdhj.2021.03.001.
3. Hendriks, T. *et al.* Effect of Systolic Blood Pressure on Left Ventricular Structure and Function: A Mendelian Randomization Study. *Hypertension* **74**, 826–832 (2019).
4. Cuspidi, C. *et al.* Improving cardiovascular risk stratification in essential hypertensive patients by indexing left ventricular mass to height(2.7). *J Hypertens* **27**, 2465–2471 (2009).
5. Aung, N. *et al.* Genome-Wide Analysis of Left Ventricular Image-Derived Phenotypes Identifies Fourteen Loci Associated With Cardiac Morphogenesis and Heart Failure Development. *Circulation* **140**, 1318–1330 (2019).
6. Begay, R. L. *et al.* Filamin C Truncation Mutations Are Associated With Arrhythmogenic Dilated Cardiomyopathy and Changes in the Cell-Cell Adhesion Structures. *JACC Clin Electrophysiol* **4**, 504–514 (2018).
7. Valdés-Mas, R. *et al.* Mutations in filamin C cause a new form of familial hypertrophic cardiomyopathy. *Nat Commun* **5**, 5326 (2014).
8. Brodehl, A. *et al.* Mutations in FLNC are Associated with Familial Restrictive Cardiomyopathy. *Hum Mutat* **37**, 269–279 (2016).
9. Locke, A. E. *et al.* Genetic studies of body mass index yield new insights for obesity biology. *Nature* **518**, 197–206 (2015).
10. The Million Veteran Program *et al.* Genetic analysis of over 1 million people identifies 535 new loci associated with blood pressure traits. *Nat Genet* **50**, 1412–1425 (2018).
11. Mahajan, A. *et al.* Multi-ancestry genetic study of type 2 diabetes highlights the power of diverse populations for discovery and translation. *Nat Genet* **54**, 560–572 (2022).
12. Tadros, R. *et al.* Shared genetic pathways contribute to risk of hypertrophic and dilated cardiomyopathies with opposite directions of effect. *Nat Genet* **53**, 128–134 (2021).
13. Walsh, R. *et al.* Quantitative approaches to variant classification increase the yield and precision of genetic testing in Mendelian diseases: the case of hypertrophic cardiomyopathy. *Genome Med* **11**, 5 (2019).
14. Fahed, A. C. *et al.* Polygenic background modifies penetrance of monogenic variants for tier 1 genomic conditions. *Nat Commun* **11**, 3635 (2020).
15. Karczewski, K. J. *et al.* The mutational constraint spectrum quantified from variation in 141,456 humans. *Nature* **581**, 434–443 (2020).
16. McLaren, W. *et al.* The Ensembl Variant Effect Predictor. *Genome Biol* **17**, 122 (2016).
17. Jurgens, S. J. *et al.* Analysis of rare genetic variation underlying cardiometabolic diseases and traits among 200,000 individuals in the UK Biobank. *Nat Genet* **54**, 240–250 (2022).
18. Liu, X., Wu, C., Li, C. & Boerwinkle, E. dbNSFP v3.0: A One-Stop Database of Functional Predictions and Annotations for Human Nonsynonymous and Splice-Site SNVs. *Hum Mutat* **37**, 235–241 (2016).
19. Pirruccello, J. P. *et al.* Analysis of cardiac magnetic resonance imaging in 36,000 individuals yields genetic insights into dilated cardiomyopathy. *Nat Commun* **11**, 2254 (2020).
20. Loh, P.-R., Kichaev, G., Gazal, S., Schoech, A. P. & Price, A. L. Mixed-model association for biobank-scale datasets. *Nat Genet* **50**, 906–908 (2018).
21. Lloyd-Jones, L. R. *et al.* Inference on the Genetic Basis of Eye and Skin Color in an Admixed Population via Bayesian Linear Mixed Models. *Genetics* **206**, 1113–1126 (2017).
22. Caliebe, A. *et al.* Including diverse and admixed populations in genetic epidemiology research. *Genetic Epidemiology* **46**, 347–371 (2022).

23. Wojcik, G. L. *et al.* Genetic analyses of diverse populations improves discovery for complex traits. *Nature* **570**, 514–518 (2019).
24. Page, G. P. *et al.* Multiple-ancestry genome-wide association study identifies 27 loci associated with measures of hemolysis following blood storage. *Journal of Clinical Investigation* **131**, e146077 (2021).
25. Daehmlow, S. *et al.* Novel mutations in sarcomeric protein genes in dilated cardiomyopathy. *Biochem Biophys Res Commun* **298**, 116–120 (2002).
26. Watkins, H. *et al.* Mutations in the cardiac myosin binding protein-C gene on chromosome 11 cause familial hypertrophic cardiomyopathy. *Nat Genet* **11**, 434–437 (1995).
27. Ai, S. *et al.* Effects of glycemic traits on left ventricular structure and function: a mendelian randomization study. *Cardiovasc Diabetol* **21**, 109 (2022).

REVIEWER COMMENTS

Reviewer #1 (Remarks to the Author):

This revision has addressed several of my previous points but has not really addressed my main concerns of validity and significance.

1. The statement of accuracy in the Limitations section is misleading: "Second, we used a deep learning model to estimate CMR-derived LVM, and therefore imperfect accuracy of estimations may have lead to reduced functional power to detect genetic associations." The main reason the model lacks accuracy is not because it is a deep learning model but because it was trained on a "ground truth" segmentation with known quality issues and without a quality control protocol. This should be added to the Limitations section.
2. "Nevertheless, we note that the model we utilized (ML4Hseg) is accurate, having a correlation of $r=0.86$ with hand-labeled CMR-derived LVM in the UK Biobank". However, correlation is not a measure of accuracy. The 95% limits of agreement stated in [60] are -27 to 27 g, after correction for linear bias. This is relatively high compared with inter-observer manual limits of agreement of -5 to 8 g in Petersen et al DOI 10.1186/s12968-017-0327-9 and the deep learning agreement of -18 to 18g in Bai et al 10.1186/s12968-018-0471-x. This comparison of limits of agreement should be added to the Limitations section.
3. It is not clear what is the significance of the study over other GWAS studies (eg Aung 2019). The "novelty" of the 11 associations is debatable since many have been previously associated with cardiac traits and most of these are likely to effect mass. A discussion of which associations which have not been previously associated with cardiac traits should be added. The prior associations with cardiac traits should be added to Table 13.

Reviewer #2 (Remarks to the Author):

The authors provide in this revised version of the manuscript a very detailed and thoughtful response to the comments of all reviewers. This reviewer appreciates the significant efforts by the authors.

My major concerns were addressed.

One point however remains related to the inclusion of participants with DCM or HCM. The authors state that there is growing evidence that other 'polygenic' effects may contribute to the phenotype or progression of individuals with these monogenic diseases. This is certainly true as the presence of modifiers influencing monogenic disease has been well recognized for some time. The statement that many patients with HCM or DCM don't harbor known mutations doesn't refute the fact that the progression in these patients is likely determined by different and divergent modifier genes. In any case, the authors provide an analysis for these subgroups. This analysis might suggest that the PRS does not significantly contribute as determined by the 95% CI described for these subgroups. While many studies in the field of LVH research simply exclude individuals with DCM or HCM independent of the fact whether a disease mutation is present or not, the results of the subgroup analysis would suggest that the inclusion in this study will simply reduce the power of the PRS analysis. I would suggest that the authors highlight in the result or discussion section these findings briefly. This might direct and clarify the interested reader to these findings which in general could be interesting and certainly clarifies the results for this subset of participants.

Reviewer #3 (Remarks to the Author):

I thank the authors for conducting extensive analyses to sufficiently address my comments.

We thank the editors and reviewers for their comments, which we feel have strengthened the manuscript.

General revisions:

In addition to specific revisions outlined below, we have moved previous Tables 1 and 3 to the Supplementary Materials and reduced the word count in the abstract and main text to comply with journal requirements.

Reviewer #1:

R1.1. The statement of accuracy in the Limitations section is misleading: “Second, we used a deep learning model to estimate CMR-derived LVM, and therefore imperfect accuracy of estimations may have lead to reduced functional power to detect genetic associations.” The main reason the model lacks accuracy is not because it is a deep learning model but because it was trained on a "ground truth" segmentation with known quality issues and without a quality control protocol. This should be added to the Limitations section.

Author Response: In response to the reviewer’s comment we have revised the referenced statement to more clearly reflect limitations of the deep learning approach used in the current study.

Revised manuscript:

Discussion (Page 15, Line 1): Second, we used a previously published deep learning model (ML4H_{seg}) to facilitate well-powered GWAS of CMR-derived LVM. ML4H_{seg} was trained using an imperfect segmentation method as ground truth,^{1,2} which may have led to lower agreement with true LVM as compared to some alternative approaches (e.g., 95% limits of agreement -27g to 27g with ML4H_{seg} versus -18g to 18g by Bai et al. using a proprietary deep learning model³ and -5 to 8g by Petersen et al. using a smaller sample of hand-labeled measurements⁴). Nevertheless, estimates from ML4H_{seg} correlate strongly ($r=0.86$) with hand-labeled CMR-derived LVM in the UK Biobank,⁵ and MR analyses recapitulated a known causal relationship between elevated blood pressure and increased indexed LVM.⁶

R1.2. “Nevertheless, we note that the model we utilized (ML4H_{seg}) is accurate, having a correlation of $r=0.86$ with hand-labeled CMR-derived LVM in the UK Biobank”. However, correlation is not a measure of accuracy. The 95% limits of agreement stated in [60] are -27 to 27 g, after correction for linear bias. This is relatively high compared with inter-observer manual limits of agreement of -5 to 8 g in Petersen et al DOI 10.1186/s12968-017-0327-9 and the deep learning agreement of -18 to 18g in Bai et al 10.1186/s12968-018-0471-x. This comparison of limits of agreement should be added to the Limitations section.

Author Response: In response to the reviewer’s comment we have added the comparison of limits of agreement to the Limitations section.

Revised manuscript:

Discussion (Page 15, Line 3): Second, we used a previously published deep learning model (ML4H_{seg}) to facilitate well-powered GWAS of CMR-derived LVM. ML4H_{seg} was trained using an imperfect segmentation method as ground truth,^{1,2} which may have led to lower agreement with true LVM as compared to some alternative approaches (e.g., 95% limits of agreement -27g to 27g with ML4H_{seg}

versus -18g to 18g by Bai et al. using a proprietary deep learning model³ and -5 to 8g by Petersen et al. using a smaller sample of hand-labeled measurements⁴). Nevertheless, estimates from ML4H_{seg} correlate strongly ($r=0.86$) with hand-labeled CMR-derived LVM in the UK Biobank,⁵ and MR analyses recapitulated a known causal relationship between elevated blood pressure and increased indexed LVM.⁶

R1.3. It is not clear what is the significance of the study over other GWAS studies (eg Aung 2019). The “novelty” of the 11 associations is debatable since many have been previously associated with cardiac traits and most of these are likely to effect mass. A discussion of which associations which have not been previously associated with cardiac traits should be added. The prior associations with cardiac traits should be added to Table 13.

Author Response: In response to the reviewer’s comment we have added an analysis of prior associations with cardiovascular diseases and risk factors (in addition to our existing comparison with LV structural measurements). We now summarize the findings in more detail in the manuscript, and have added the requested information to **Supplementary Table 12** (previously Supplementary Table 13). To increase clarity, we now specify in greater detail which associations are novel and in what manner (i.e., novel for LVM, novel for structural traits, and novel for any cardiovascular disease or risk factor). Although the reviewer is correct that many of the loci we identified have prior associations with cardiac traits (e.g., atrial fibrillation), we submit that identification of associations with LVM are nevertheless valuable as they may provide further insight into potential mechanisms underlying those prior associations (e.g., greater LVM may lead to greater risk of atrial fibrillation). Interestingly, we also note that a number of loci we identified are novel for LVM but have prior associations with electrocardiographic traits. We submit future work is warranted to assess whether such associations may reflect electrical manifestations of LV mass or the presence of a cardiomyopathy.

Revised manuscript:

Abstract (Page 3, Line 7): We identify 12 genome-wide associations (1 known at *TTN* and 11 novel for LVM), implicating genes previously associated with cardiac contractility and cardiomyopathy.

Methods (Page 22, Line 10): To assess whether the variants we identified in association with LVMI have been previously associated with other LV measurements, we compared our loci to those reported to have genome-wide associations with other LV measurements in prior analyses by Pirruccello et al.⁷ and Aung et al.⁸ We performed an analogous search for associations with any cardiovascular disease or risk factor using the National Human Genome Research Institute GWAS Catalog. For these analyses, we tabulated all associations including the same variant, a variant serving as a strong proxy ($r^2 \geq 0.80$), or a variant mapping to the same candidate gene.

Results (Page 8, Line 18): We assessed whether the significant loci we identified have been previously associated with LV measurements^{7,8} and cardiovascular traits. Including the European-only analysis, a total of 4 loci have been previously associated with LV measurements. Variant rs2255167 is located on a region of *TTN* previously associated with LV mass, LV end diastolic volume, LV end systolic volume, and LV ejection fraction. Variants rs6503451 near *MAPT* and rs199501 near *WNT3* are located at regions previously associated with LV end-systolic volume. In the European-only analysis, variant rs143973349 near *FLNC* is at a locus previously associated with LV end-systolic volume and LV

ejection fraction. Several additional loci have been implicated in other cardiovascular diseases such as heart failure (e.g., rs34163229 near *SYNPO2L*), cardiomyopathy (e.g., rs2255167 near *TTN*, rs3729989 near *MYBPC3*, rs143973349 near *FLNC*), and atrial fibrillation (e.g., rs6598541 near *IGF1R*), while others have been associated with cardiovascular risk factors such as blood pressure or diabetes. Several variants are located at regions previously associated with electrocardiographic traits such as PR interval (e.g., rs56252725 near *PDXDC1*), QRS duration (rs6598541 near *IGF1R*), and QRS amplitude (rs6503451 near *MAPT*). Variants rs28552516 near *KDM2B*, rs62621197 near *ADAMTS10*, and rs142032045 near *DOC2A* in the European-only analysis have not been previously associated with either LV or other cardiovascular traits. A summary of lead variants and their prior associations is shown in

Supplementary Table 13.

Discussion (Page 11, Line 14): Leveraging favorable statistical power and a rich imaging-based phenotype, we identified 12 independent loci associated with LVMI at genome-wide significance. Of the loci identified, 11 are novel for LV mass, 9 have not been previously associated with any LV measurement, and 2 have not been associated with any cardiovascular trait or risk factor. A European-only analysis revealed 2 additional loci which are novel for LV mass.

Discussion (Page 13, Line 21): Interestingly, we identified several loci which are novel for LVM but have prior associations with electrocardiographic traits.^{9,10} Future work is warranted to assess whether such associations may reflect electrical manifestations of LV mass or the presence of a cardiomyopathy.

Supplementary Table 12. Prior associations of lead variants with CMR-derived left ventricular measurements and cardiovascular traits

rsID	Chr	Position (hg38)	Prioritized candidate gene(s)	Beta (SE)	P-value	Prior associations with left ventricular traits*	Prior associations with cardiovascular traits*
Primary association analysis							
rs143800963	1	11835418	CLCN6 NPPA NPPB	0.95 (0.16)	4.2x10 ⁻⁹		Systolic BP (rs6669371 ¹¹), NTproBNP levels (rs1023252 ¹²)
rs2255167	2	178693555	TTN FKBP7	0.97 (0.09)	3.2x10 ⁻²⁶	LVM (rs225516 ⁸) LVEDV (rs2042995 ⁸) LVESV (rs2042995 ⁸ , rs2562845 ⁷) LVESVi (rs2562845 ⁷) LVEF (rs2042995 ⁸ , rs2562845 ⁷)	DCM (rare variants ¹³), HCM (rare variants ¹⁴), atrial fibrillation (rare variants ¹⁵)
rs10497529	2	178975161	TTN CCDC141	1.28 (0.20)	2.2x10 ⁻⁹		Resting heart rate (rs10497529 ¹⁶), peak oxygen consumption (rs10497529 ¹⁷)
-	5	133066736	HSPA4	0.50 (0.08)	1.6x10 ⁻⁹		Systolic BP (rs62374461 ¹⁸), diastolic BP (rs55747751 ¹⁹)
rs9388498	6	126552277	CENPW	-0.55 (0.10)	4.1x10 ⁻⁹		Coronary artery disease (rs1591805 ²⁰), Type 1 diabetes (rs1538171 ²¹), Type 2 diabetes (rs4897182 ²²)
rs34163229	10	73647154	SYNPO2L MYOZ1 ANXA7 PPP3CB AGAP5	-0.60 (0.10)	1.0x10 ⁻⁸		Systolic BP (rs12247028 ²³), atrial fibrillation (multiple ²⁴), heart failure (rs4746140 ²⁵), QT interval (rs4746140 ²⁶), PR interval (rs7394152 ¹⁰)
rs3729989	11	47348490	MYBPC3 PSMC3	-0.61 (0.11)	1.8x10 ⁻⁸		Systolic BP (rs2301216 ¹⁸), renin-angiotensin system inhibitor use (rs2856653 ²⁷), HCM (rare variants ²⁸), DCM (rare variants ²⁹)

rs28552516	12	121592356	ORAI1	-0.58 (0.10)	1.5x10 ⁻⁸		
rs6598541	15	98727906	IGF1R	-0.42 (0.08)	4.6x10 ⁻⁸		TPE interval (rs2871974 ³⁰), QRS duration (rs4966020 ³¹), atrial fibrillation (rs4965430 ³²)
rs56252725	16	14995819	PDXDC1 NOMO1 MYH11	0.54 (0.09)	3.7x10 ⁻⁹		Coronary artery disease (rs216158 ³³), PR interval (rs72772025 ¹⁰), resting heart rate (rs3915499 ³⁴), systolic BP (rs3915499 ¹⁸), ECG morphology (rs3915425 ⁹)
rs6503451	17	45870981	KANSL1 MAPT LRRC37A LRRC37A2	-0.52 (0.08)	1.1x10 ⁻¹⁰	LVESV (rs242562 ⁸) LVESVi (rs242562 ⁸)	Systolic BP (rs3785880 ¹⁸), QRS amplitude (rs242562) ³⁵
rs199501	17	46785247	WNT3 KANSL1 LRRC37A LRRC37A2	0.55 (0.09)	1.1x10 ⁻⁹	LVESV (rs242562 ⁸) LVESVi (rs242562 ⁸)	Atrial fibrillation (rs1563304 ³²)
rs62621197	19	8605262	ADAMTS10 MYO1F	1.11 (0.20)	2.9x10 ⁻⁸		
European ancestry analysis							
rs143973349	7	128866181	FLNC CCDC136 KCP	0.57 (0.10)	1.0x10 ⁻⁰⁸	LVESV (rs34373805 ⁸) LVESVi (rs34373805 ⁸) LVEF (rs3807309 ⁸)	ECG morphology (rs56216811 ⁹), arrhythmogenic CM (rare variants ³⁶), HCM (rare variants ³⁷), restrictive CM (rare variants ³⁸)
rs142032045	16	30018280	DOC2A	-0.45 (0.08)	3.9x10 ⁻⁰⁸		
*Traits listed for prior associations with the respective lead variant, a strong proxy ($r^2 \geq 0.80$), or a variant mapped to the same gene. The variant implicated in the prior analysis is listed in parenthesis. BP = blood pressure, Chr = chromosome; CM = cardiomyopathy, DCM = dilated cardiomyopathy, ECG = electrocardiogram, HCM = hypertrophic cardiomyopathy, LVEF = left ventricular ejection fraction; LVESV = left ventricular end-systolic volume; LVESVi = left ventricular end-systolic volume index, SE = standard error, TPE = T wave peak-to-end							

Reviewer #2:

R2.1. The authors provide in this revised version of the manuscript a very detailed and thoughtful response to the comments of all reviewers. This reviewer appreciates the significant efforts by the authors.

My major concerns were addressed.

Author Response: We thank the reviewer for their thoughtful review and constructive feedback, which have improved the manuscript.

R2.2. One point however remains related to the inclusion of participants with DCM or HCM. The authors state that there is growing evidence that other 'polygenic' effects may contribute to the phenotype or progression of individuals with these monogenic diseases. This is certainly true as the presence of modifiers influencing monogenic disease has been well recognized for some time. The statement that many patients with HCM or DCM don't harbor known mutations doesn't refute the fact that the progression in these patients is likely determined by different and divergent modifier genes. In any case, the authors provide an analysis for these subgroups. This analysis might suggest that the PRS does not significantly contribute as determined by the 95% CI described for these subgroups. While many studies in the field of LVH research simply exclude individuals with DCM or HCM independent of the fact whether a disease mutation is present or not, the results of the subgroup analysis would suggest that the inclusion in this study will simply reduce the power of the PRS analysis. I would suggest that the authors highlight in the result or discussion section these findings briefly. This might direct and clarify the interested

reader to these findings which in general could be interesting and certainly clarifies the results for this subset of participants.

Author Response: We agree with the reviewer that further clarification of our approach and contextualization of our PRS findings with regard to HCM and DCM would be useful. We have added suggested discussion around these points to the manuscript.

Discussion (Page 14, Line 11): Of note, we did not exclude individuals with DCM or HCM from our analyses of incident disease since we hypothesized that polygenic risk may nevertheless contribute to the development of clinical outcomes.³⁹ In the context of low event rates, however, the LVMI PRS was associated with incident DCM only in the UK Biobank, and associations with incident HCM were not significant in either sample.

Reviewer #3:

R3.1. I thank the authors for conducting extensive analyses to sufficiently address my comments.

Author Response: We thank the reviewer for their thoughtful review and constructive feedback, which have improved the manuscript.

References

1. Khurshid, S. *et al.* Deep learning to estimate cardiac magnetic resonance–derived left ventricular mass. *Cardiovascular Digital Health Journal* S2666693621000232 (2021) doi:10.1016/j.cvdhj.2021.03.001.
2. Suinesiaputra, A. *et al.* Fully-automated left ventricular mass and volume MRI analysis in the UK Biobank population cohort: evaluation of initial results. *Int J Cardiovasc Imaging* **34**, 281–291 (2018).
3. Bai, W. *et al.* Automated cardiovascular magnetic resonance image analysis with fully convolutional networks. *J Cardiovasc Magn Reson* **20**, 65 (2018).
4. Petersen, S. E. *et al.* Reference ranges for cardiac structure and function using cardiovascular magnetic resonance (CMR) in Caucasians from the UK Biobank population cohort. *J Cardiovasc Magn Reson* **19**, 18 (2017).
5. Khurshid, S. *et al.* Deep learning to estimate cardiac magnetic resonance–derived left ventricular mass. *Cardiovascular Digital Health Journal* S2666693621000232 (2021) doi:10.1016/j.cvdhj.2021.03.001.
6. Hendriks, T. *et al.* Effect of Systolic Blood Pressure on Left Ventricular Structure and Function: A Mendelian Randomization Study. *Hypertension* **74**, 826–832 (2019).
7. Pirruccello, J. P. *et al.* Analysis of cardiac magnetic resonance imaging in 36,000 individuals yields genetic insights into dilated cardiomyopathy. *Nat Commun* **11**, 2254 (2020).
8. Aung, N. *et al.* Genome-Wide Analysis of Left Ventricular Image-Derived Phenotypes Identifies Fourteen Loci Associated With Cardiac Morphogenesis and Heart Failure Development. *Circulation* **140**, 1318–1330 (2019).
9. Verweij, N. *et al.* The Genetic Makeup of the Electrocardiogram. *Cell Syst* **11**, 229-238.e5 (2020).
10. Ntalla, I. *et al.* Multi-ancestry GWAS of the electrocardiographic PR interval identifies 202 loci underlying cardiac conduction. *Nat Commun* **11**, 2542 (2020).
11. Giri, A. *et al.* Trans-ethnic association study of blood pressure determinants in over 750,000 individuals. *Nat Genet* **51**, 51–62 (2019).
12. Del Greco M, F. *et al.* Genome-wide association analysis and fine mapping of NT-proBNP level provide novel insight into the role of the MTHFR-CLCN6-NPPA-NPPB gene cluster. *Hum Mol Genet* **20**, 1660–1671 (2011).
13. Herman, D. S. *et al.* Truncations of titin causing dilated cardiomyopathy. *N Engl J Med* **366**, 619–628 (2012).
14. Satoh, M. *et al.* Structural analysis of the titin gene in hypertrophic cardiomyopathy: identification of a novel disease gene. *Biochem Biophys Res Commun* **262**, 411–417 (1999).
15. Choi, S. H. *et al.* Association Between Titin Loss-of-Function Variants and Early-Onset Atrial Fibrillation. *JAMA* **320**, 2354–2364 (2018).
16. Eppinga, R. N. *et al.* Identification of genomic loci associated with resting heart rate and shared genetic predictors with all-cause mortality. *Nat Genet* **48**, 1557–1563 (2016).
17. Hanscombe, K. B. *et al.* The genetic case for cardiorespiratory fitness as a clinical vital sign and the routine prescription of physical activity in healthcare. *Genome Med* **13**, 180 (2021).
18. Kichaev, G. *et al.* Leveraging Polygenic Functional Enrichment to Improve GWAS Power. *Am J Hum Genet* **104**, 65–75 (2019).
19. the Million Veteran Program *et al.* Genetic analysis of over 1 million people identifies 535 new loci associated with blood pressure traits. *Nat Genet* **50**, 1412–1425 (2018).
20. van der Harst, P. & Verweij, N. Identification of 64 Novel Genetic Loci Provides an Expanded View on the Genetic Architecture of Coronary Artery Disease. *Circ Res* **122**, 433–443 (2018).

21. Onengut-Gumuscu, S. *et al.* Fine mapping of type 1 diabetes susceptibility loci and evidence for colocalization of causal variants with lymphoid gene enhancers. *Nat Genet* **47**, 381–386 (2015).
22. Zhao, W. *et al.* Identification of new susceptibility loci for type 2 diabetes and shared etiological pathways with coronary heart disease. *Nat Genet* **49**, 1450–1457 (2017).
23. CHARGE-EchoGen Consortium *et al.* The genetics of blood pressure regulation and its target organs from association studies in 342,415 individuals. *Nat Genet* **48**, 1171–1184 (2016).
24. Roselli, C. *et al.* Multi-ethnic genome-wide association study for atrial fibrillation. *Nat Genet* **50**, 1225–1233 (2018).
25. Shah, S. *et al.* Genome-wide association and Mendelian randomisation analysis provide insights into the pathogenesis of heart failure. *Nat Commun* **11**, 163 (2020).
26. Young, W. J. *et al.* Genetic analyses of the electrocardiographic QT interval and its components identify additional loci and pathways. *Nat Commun* **13**, 5144 (2022).
27. Wu, Y. *et al.* Genome-wide association study of medication-use and associated disease in the UK Biobank. *Nat Commun* **10**, 1891 (2019).
28. Watkins, H. *et al.* Mutations in the cardiac myosin binding protein-C gene on chromosome 11 cause familial hypertrophic cardiomyopathy. *Nat Genet* **11**, 434–437 (1995).
29. Daehmlow, S. *et al.* Novel mutations in sarcomeric protein genes in dilated cardiomyopathy. *Biochem Biophys Res Commun* **298**, 116–120 (2002).
30. Ramírez, J. *et al.* Common Genetic Variants Modulate the Electrocardiographic Tpeak-to-Tend Interval. *Am J Hum Genet* **106**, 764–778 (2020).
31. Prins, B. P. *et al.* Exome-chip meta-analysis identifies novel loci associated with cardiac conduction, including ADAMTS6. *Genome Biol* **19**, 87 (2018).
32. Nielsen, J. B. *et al.* Biobank-driven genomic discovery yields new insight into atrial fibrillation biology. *Nat Genet* **50**, 1234–1239 (2018).
33. Koyama, S. *et al.* Population-specific and trans-ancestry genome-wide analyses identify distinct and shared genetic risk loci for coronary artery disease. *Nat Genet* **52**, 1169–1177 (2020).
34. Ramírez, J. *et al.* Thirty loci identified for heart rate response to exercise and recovery implicate autonomic nervous system. *Nat Commun* **9**, 1947 (2018).
35. van der Harst, P. *et al.* 52 Genetic Loci Influencing Myocardial Mass. *J Am Coll Cardiol* **68**, 1435–1448 (2016).
36. Begay, R. L. *et al.* Filamin C Truncation Mutations Are Associated With Arrhythmogenic Dilated Cardiomyopathy and Changes in the Cell-Cell Adhesion Structures. *JACC Clin Electrophysiol* **4**, 504–514 (2018).
37. Valdés-Mas, R. *et al.* Mutations in filamin C cause a new form of familial hypertrophic cardiomyopathy. *Nat Commun* **5**, 5326 (2014).
38. Brodehl, A. *et al.* Mutations in FLNC are Associated with Familial Restrictive Cardiomyopathy. *Hum Mutat* **37**, 269–279 (2016).
39. Fahed, A. C. *et al.* Polygenic background modifies penetrance of monogenic variants for tier 1 genomic conditions. *Nat Commun* **11**, 3635 (2020).

REVIEWERS' COMMENTS

Reviewer #1 (Remarks to the Author):

All my concerns were addressed in this revision.

We thank the editors and reviewers for their comments, which we feel have strengthened the manuscript.

Editors:

We have revised the manuscript in accordance with editorial requests, as described in detail in the revised author checklist.

Reviewer #1:

R1.1. All my concerns were addressed in this revision.

Author Response: We thank the reviewer for their thoughtful review and constructive feedback, which we feel have improved the manuscript.